# Lagrangian Modelling of a Person lost at Sea during Adriatic Scirocco Storm of 29 October 2018

Matjaž Ličer[1], Solène Estival[2], Catalina Reyes-Suarez[3], Davide Deponte[3], and Anja Fettich[4]

[1]National Institute of Biology, Piran, Slovenia
[2]École Nationale Supérieure de Techniques Avancées, Paris, France
[3]Istituto Nazionale di Oceanografia e Geofisica Sperimentale, Sgonico, Italy
[4]Slovenian Environment Agency, Ljubljana, Slovenia

**Correspondence:** Matjaž Ličer (matjaz.licer@nib.si)

**Abstract.** On 29 October 2018 a windsurfer's mast broke about 1 km offshore from Istria during a severe Scirocco storm in the Northern Adriatic Sea. He drifted in severe marine conditions until he eventually beached alive and well in Sistiana (Italy) 24 hours later. We conducted an interview with the survivor to reconstruct his trajectory and to gain insight into his swimming and paddling strategy. Part of survivor's trajectory was verified using high-frequency radar surface current observations as inputs for Lagrangian temporal back-propagation from the beaching site. Back-propagation simulations were found to be largely consistent with survivor's reconstruction. We then attempted a Lagrangian forward-propagation simulation of his trajectory by performing a leeway simulation using the OpenDrift tracking code using two object types: i) person in water in unknown state and ii) person with surf board. In both cases a high-resolution (1 km) setup of NEMO v3.6 circulation model was employed for the surface current component and a 4.4 km operational setup of the ALADIN atmospheric model was used for wind forcing. Best performance is obtained using "person with surfboard" object type, giving the highest percentage of particles stranded within 5 km of the beaching site. Accumulation of particles stranded within 5 km of the beaching site saturates 6 hours after the actual beaching time for all drifting particle types. This time-lag most likely occurs due to poor NEMO model representation of surface currents in the final hours of the drift. A control run of wind-only forcing shows the poorest performance of all simulations. This indicates the importance of topographically constrained ocean currents in semi-enclosed basins even in seemingly wind-dominated situations for determining the trajectory of a person lost at sea.

## 1 Introduction

Lagrangian particle tracking of objects lost at sea is an important branch of ocean forecasting. Maritime search and rescue (SAR) or other types of civil service responses depend on timely and reliable estimates of the most probable areas which contain the drifting object. These estimates generally require prior computation of ocean currents, waves and winds in the area, which are most often provided by numerical circulation, wave and atmosphere models.

The wind force contribution to the objects drift is termed its leeway and has both downwind (drag) and crosswind (lift) component (Breivik and Allen, 2008). The object's drift therefore generally deviates from the wind direction by some divergence angle $L_\alpha$ (Allen and Plourde, 1999), related to the downwind and crosswind components. Specific values of the object's down-

wind and crosswind drift are determined by the balance of the wind (lift and drag) force on the overwater part of the object and the hydrodynamic (lift and drag) force on the subsurface part of the object - object's drifting properties therefore depend significantly on its shape. Empirical observations have consequently been the most straightforward method of determining the drifting parameters for various drifting object types, including human bodies (Allen and Plourde, 1999; Hackett et al., 2006). Reports on marine drifts involving survivors are not ubiquitous, which makes reviews like the one from Allen and Plourde (1999) all the more valuable for modelling marine drifts of persons or other objects.

In this paper we focus on an incident which occured on 29 October 2018 in the Northern Adriatic Sea and led to a 24 hour drift of a person in gale wind conditions (level 8 on Beaufort scale). For an extensive analysis of the atmospheric and marine conditions during the 29 October 2018 storm the reader is referred to Cavaleri et al. (2019). These conditions are related to the fact that the Adriatic sea is a northwest-southeast oriented elongated basin of the Northern Central Mediterranean, exchanging properties with the eastern Mediterranean basin through the Otranto strait (19° E, 40° N in Figure 1 a) ). It is 800 km long and 200 km wide and surrounded from all sides by mountain ridges - the Alps in the north, the Apennines in the west and Dinaric Alps in the east. These ridges exhibit significant influence on the basin circulation through topographic control of the air flow, most notably during episodes of the northeasterly Bora wind and southeasterly Scirocco. The northern part of the Adriatic is a shallow shelf with depths under 60 m. Its northernmost part, extending into the Gulf of Trieste, is the shallowest, with depths around 20 to 30 m (see Figure 1 b) ).

In the afternoon of 29 Oct 2018, the Scirocco speeds along the west coast of northern Istria were in the range 15-25 m s$^{-1}$ and significant wave heights amounted to 3-5 m (Cavaleri et al., 2019), while maximum wave heights in the southern part of the Gulf of Trieste at coastal buoy Vida (see Section 2.1 for details and Figure 1 b) for location) were observed to be over 2.5 m (not shown). The town of Umag in northern Istria is a popular windsurfing spot during Scirocco conditions: on 29 Oct 2018 many people were windsurfing there when the accident occured at estimated 16 UTC. The windsurfer's mast broke roughly 1 km offshore northwest of Umag (see Figure 1 b) for location) initiating the drift. The conditions were too severe for immediate marine rescue either by his colleagues or by authorities. A joint Italian, Croatian and Slovenian SAR mission was initiated next morning (30 Oct 2018) but it was unsuccesful - the surfer beached on his own 24 hours later close to Sistiana north of Trieste (see Figure 1 b) ). The windsurfer's harness was however recovered in the central part of the Gulf of Trieste at around 15 UTC on 30 Oct.

The survivor kindly responded to our interview request. He is an experienced windsurfer and has been windsurfing along the coasts of Gulf of Trieste for the past 30 years. We state that explicitly to convey the fact that he knows this coastline very well. We now briefly recapitulate his personal statements about the drift. He was conscious and focused the entire time. The visibility was not bad and he could see the coastline of the Gulf of Trieste in its entirety, which helped him make mental notes of his location. He was highly alert to his location throughout the drift but did not have a watch or a GPS. We have therefore attempted to independently validate his trajectory estimate, as will be explained below in Section 4.1.

His mast broke on 29 Oct 2019 16 UTC at 13.625° E, 45.558° N with an estimated ± 500 m error in each direction, see Figure 1 b) for location. Immediately after the accident, he drifts alongshore north of Umag and he actively paddles towards the coast, hoping to reach the Cape of Savudrija. The wind direction at his location is however slightly offshore and sometime

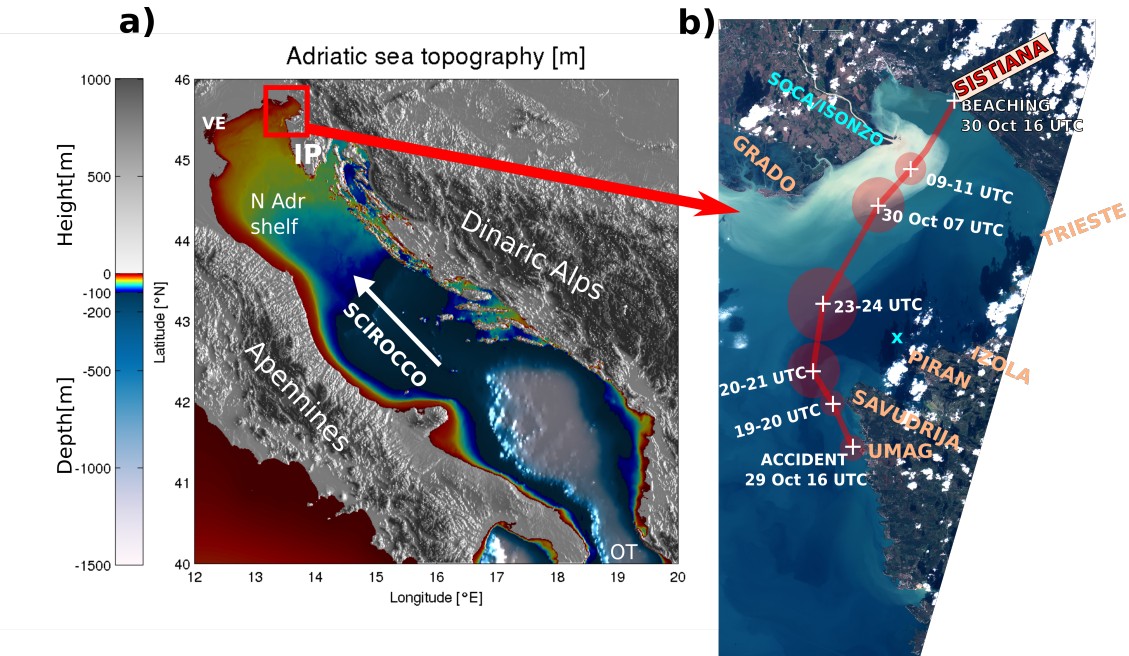

**Figure 1.** a) Adriatic basin bathymetry. Abbreviations are as follows: VE - Venice, IP - Istrian Peninsula, N Adr Shelf - Northern Adriatic Shelf, OT - Otranto Strait. Direction of Scirocco is marked with white arrow. b) The Gulf of Trieste and piecewise trajectory of the drift as estimated by the survivor. Location estimates are junctions of the piecewise straight line. Circles denote location uncertainty estimates at specific times. The cyan 'x' sign north of Piran denotes the location of the Vida coastal buoy. Background layer is Sentinel-2 L1C True Color image of the Gulf of Trieste from the day after the beaching, 31 Oct 2018 (obtained from Copernicus Open Access Hub: https://scihub.copernicus.eu). Turbid Soča/Isonzo river plume is clearly visible along the northern shore of the Gulf.

between 19.30 and 20.30 UTC he realizes he will not be able to reach Savudrija. After 20 UTC the Scirocco strengthens. He
is now located northwest of Savudrija, drifting north-northwest toward Grado. Swimming is not possible due to airspray and
sea conditions, but he keeps shaking his arms and legs interchangeably to keep warm. At some point between 20 UTC and
23 UTC he can see the town of Izola (Slovenia) and the town of Grado (Italy) at right angles. It is around 23 UTC that his
drift turns north-east. After 23 UTC, he is located approximately on the Piran-Grado line. Sea conditions get very severe, he is
laying on the windsurf board, mostly facing southwest, away from the mean drift direction, drifting backwards, clutching the
footstraps on the surfboard. He estimates that every 50th wave breaks over him and pulls the surfboard from under him. When
this happens he needs to wait to reach the crest of the wave to visually re-locate the board and catch it. In the morning, on 30
Oct 2019 07 UTC, he is located 2 - 4 km south-southwest of the Soča/Isonzo river mouth. By 9-10 UTC he is located roughly
1-2 km south-southeast of the river mouth and the water gets significantly colder as he likely enters the Soča/Isonzo river
plume (visible in Figure 1 b) ). By the time he is entering the plume, the Soča/Isonzo runoff is at a several-month maximum,
as depicted in Figure 2. From 11 UTC on he is paddling actively toward northeast to overcome the riverine westward coastal
current until he reaches the beach near Sistiana at 16 UTC.

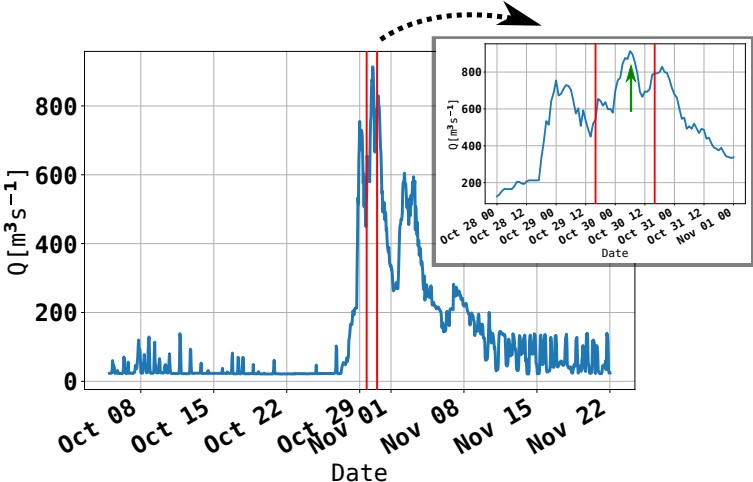

**Figure 2.** Soča/Isonzo runoff during October and November 2018, as measured at an upstream river gauge (operated by ARSO) at Solkan, Slovenia. Vertical red lines indicate the timewindow of the drift. Green arrow in the inset marks approximate time of windsurfer's entering the river plume.

The drifting trajectory, reconstructed from above, is shown in the b) panel in Figure 1. Due to the nature of the testimony and
lack of measuring equipment, survivor's trajectory is burdened with error. Survivor estimated the errors in his spatiotemporal
location to best of his ability: these estimates, arbitrary as they are, are presented as semi-transparent circles around each
marked location in Figure 1 and other figures. We have however attempted to verify the final part of his trajectory by using
high-frequency radar surface current measurements (see section 4.1). HF measurements do not cover the entire area of the
drift, but they do cover the final half of the drift. We could therefore not use HF observations to force the survivor's drift from
start to finish, but we could use them to perform Lagrangian back-propagation of particles from his beaching location back
into the Gulf. As will be shown later, these results are consistent with survivors trajectory estimate. While not allowing for
any meaningful quantitative verification of the Lagrangian codes in this paper, we believe that the trajectory is a qualitatively
suitable guide for our simulations.
In the present paper, we present two attempts to simulate this trajectory using state-of-the-art particle tracking model Open-
Drift. Available observations and general marine conditions during the drift are presented in Section 2; numerical models used
for particle tracking modelling chain are described in Section 3. Lagrangian model OpenDrift and its setup is presented in
Section 3.2. Simulation results are depicted and discussed in Section 4, followed by concluding remarks in Section 5.

## 2 Observations

### 2.1 Coastal buoy Vida

The oceanographic buoy Vida is a coastal observation platform, operated by the Marine Biology Station at the National Institute of Biology (NIB). It is located in the southern part of the Gulf of Trieste at (13.55505 E, 45.5488 N), see b) panel of Figure 1 (marked with a cyan cross). Data from the buoy are multifaceted (air temperature, air humidity, currents, waves, sea temperature, salinity, dissolved oxygen, chlorophyll concentration, etc.) and are publicly available (http://www.nib.si/mbp/en/buoy/) in near real time. Ocean currents are acquired by a Nortek AWAC acoustic Doppler current profiler, mounted on the sea bottom at a depth of 22.5 m, to monitor vertical current profiles (at 1 m intervals along the water column). The top most cell of the ADCP measurement corresponds to a depth around 0.5 m. Further information on the buoy can be found in Malačič (2019).

### 2.2 High Frequncy Radar System

The HF systems deployed in the Gulf of Trieste consist of two WERA stations (Gurgel et al., 1999) manufactured by Helzel MessTechnik in Germany, one at the OGS facility in Aurisina (Italy) and the second, operated by NIB, in the urban area of Piran (Slovenia). The systems provide sea surface current maps since January 2015. They rely on the scattering of a short-duration (9 minutes) and low-power (below 20 Watts) harmless radio wave pulses from waves at the ocean surface satisfying the Bragg-resonance scattering condition for coherent return. The two systems operate at a carrier frequency of 25.5 MHz as regulated by the International Telecommunication Union, covering the Gulf of Trieste at 1 km range resolution and 1° angular resolution every 30 minutes. After aquisition, data are processed and radial components of the surface current field are obtained, which in turn are combined into a 1.5 km horizontal resolution 22×20 regular grid (see Figure 3 for coverage during the event and both station locations). Combined data are stored in databases and can be visualized in near real time at http://www.nib.si/mbp/en/oceanographic-data-and-measurements/other-oceanographic-data/hf-radar-2. WERA system external antenna field calibration was performed in 2016 and WERA system intrinsic estimates of zonal and meridional current errors amount to 1-3 $\mathrm{cm\,s^{-1}}$ (roughly 3 - 10% of observed current speed) during the period of the drift. Data availability during the 24 hours of the drift was between 50 and 70 percent, as depicted in Figure 3. The strip without data, visible in Figure 3 along the connecting line between the two WERA systems, occurs due to geometric dilution of precision: along the connecting line, the two radar stations measure exactly the same radial current components from which no reconstruction of transverse current components can be made. Lacking transverse information, no estimate of the total current can be made along this line. The two WERA HF systems are operated and maintained in collaboration between researchers, engineers and technicians from OGS and NIB.

## 3 Models

### 3.1 Ocean and Atmospheric Models

#### 3.1.1 NEMO Circulation Model

We are using a high horizontal resolution ($1°/111$ or roughly 1000 m) setup of NEMO v3.6 (Madec, 2008) over the Adriatic basin on a regular $999 \times 777$ longitude-latitude grid and 33 vertical $z^*$-levels with partial step. Model domain spans $12° - 21°$ E and $39° - 46°$ N, see Figure 3. Maximum vertical discretization stretch is located at 15th level to allow for appropriate vertical resolution near the surface. In all regions shallower than 2 m, a minimum 2 m depth is enforced. Vertical level depths in meters are 0.50, 1.51, 2.55, 3.64, 4.83, 6.20, 7.94, 10.38, 14.18, 20.56, 31.68, 51.23, 84.58, 137.94, 215.83, 318.24, 440.67, 576.90, 721.55, 870.95, 1022.92, 1176.25, 1330.29, 1484.69,1639.28, 1793.97, 1948.71, 2103.47, 2258.25, 2413.03, 2567.81, 2722.60, 2877.39. Explicit time-splitting is enforced and barotropic timestep is automatically adjusted to meet Courant-Friedrichs-Lewy stability criterion. Baroclinic timestep was set to 120 s. The model is running daily at Slovenian Environment Agency (ARSO). It is initialized from previous operational run. Hourly lateral boundary conditions in the Ionian Sea are taken from the Copernicus CMEMS MFS model. Turbulent heat and momentum fluxes across the ocean surface are computed with CORE bulk flux formulation (Large and Yeager, 2004) using ALADIN SI atmospheric fields (surface wind, cloud cover, mean sea level pressure, 2m temperature, relative humidity and precipitation). Rivers are implemented as freshwater release over the entire water column at the discharge location, with runoff values as described in Ličer et al. (2016). Tides are included as lateral boundary conditions for open boundary elevations and barotropic velocities for K1, P1, O1, Q1, M2, K2, N2 and S2 constituents. Constituents at the open boundary are obtained using OTIS tidal inversion code (Egbert and Erofeeva, 2002), based on TPXO8 atlas. The model employs Flather boundary condition for barotropic dynamics and Flow Relaxation Scheme (Engedahl, 1995) for baroclinic dynamics and tracers at the open boundary. Lateral momentum boundary condition at the coast is free-slip. Bottom friction is nonlinear with a logarithmic boundary layer. Lateral diffusion schemes for tracers and momentum are both bilaplacian over geopotential surfaces. Vertical diffusion is computed using Generic Length Scale (GLS) turbulence scheme. Craig and Banner formulation (Craig and Banner, 1994) of surface mixing due to wave breaking is switched on. Present NEMO setup does not have data assimilation and all the simulations in this paper consequently lack any data assimilation as well.

#### 3.1.2 ALADIN Atmospheric Model

The version of the model used for the experiments in this paper is currently operational at the Slovenian Weather Service. It runs on a $432 \times 432$ horizontal Lambert conic conformal grid with 4.4 km resolution and 87 vertical levels with the model top at 1 hPa and model integration time step of 180 s. The model domain spans $[0.7°\,\text{W}, 28.6°\,E]$ in longitude and $[37.4°\,\text{N},$ $55.0°\,\text{N}]$ in latitude, see Figure 3. The physics package used in the model is the so-called ALARO-0, that uses Modular, Multi-scale, Microphysics and Transport (3MT) structure (Gerard et al., 2009). Initial conditions for the model are provided by atmospheric analysis with 3 hourly three-dimensional variational assimilation (3D-Var) (Fischer et al., 2005; Strajnar et al.,

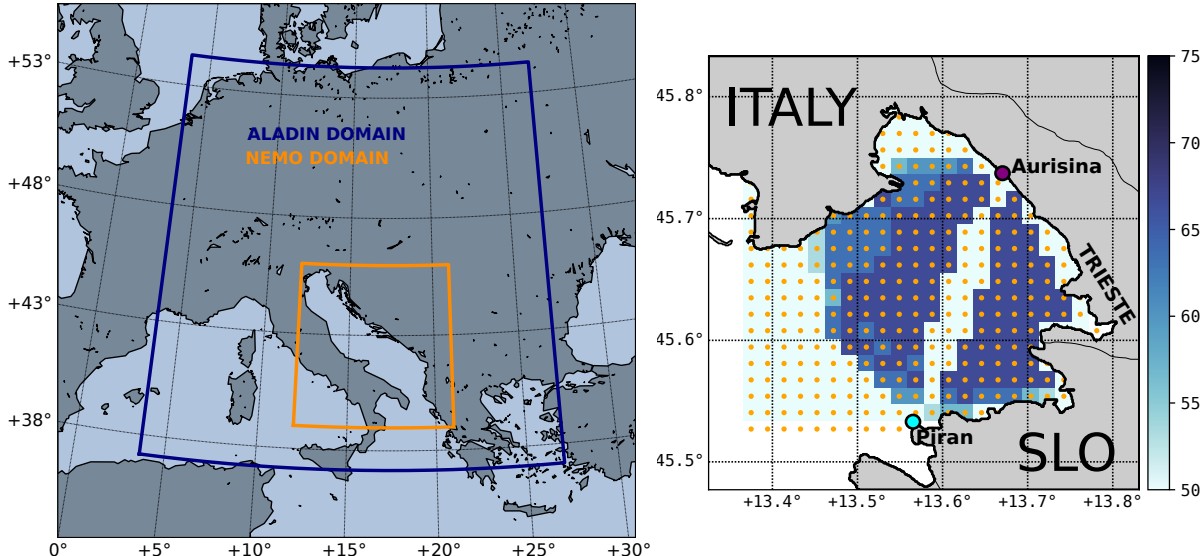

**Figure 3.** Left: Computational domains of ALADIN SI (blue) and NEMO (orange) numerical models. Right: WERA HF radar grid (orange dots) and data availability percentage per grid point between 29 Oct 2018 16 UTC and 30 Oct 2018 21 UTC.

2015) and optimal interpolation for surface and soil variables. Sea surface temperature (SST) in the model is initialized from
the most recent host model analysis of the ECMWF model that uses Operational Sea Surface Temperature and Sea Ice Analysis
(OSTIA, Donlon et al., 2012), supplied by the National Environmental Satellite, Data and Information Service (NESDIS) of
the American National Ocean and Atmospheric Administration (NOAA). Information at the domain edge is obtained from the
global model by applying Davies relaxation (Fischer et al., 1976). Lateral boundary conditions are provided by the ECMWF
Boundary Conditions Optional project and are applied with a 1 h period in the assimilation cycle and a 3 h period during model
forecasts. Boundary condition information is interpolated linearly for time steps between these times. Further details about the
model setup and assimilation scheme are available in Strajnar et al. (2015, 2019); Ličer et al. (2016).

## 3.2 Lagrangian Models and OpenDrift Setup

Lagrangian or particle tracking models are used for general purpose tracking problems from marine oil-spill dispersion mod-
elling to water age, marine bacterial transport and object drift forecasting. Typically an arbitrary number of particles $N_p$ (*i.e.*
several thousand) are seeded at the initial location and subjected in each timestep to advection, turbulent diffusion and, if appli-
cable, fate. Throughout this paper all the particles are considered passive, *i.e.* their advection is solely due to external forcing
from wind and sea. Lagrangian trajectory $\mathbf{r}_p(t)$ of $p-$th particle ($p = 1, \ldots, N_p$) is computed using a suitable numerical method
(*i.e.* Runge-Kutta or Euler method) to integrate the following initial value problem

$$\frac{d\mathbf{r}_p(t)}{dt} \quad = \quad \mathbf{u}_c(\mathbf{r}_p(t), t) + \mathbf{l}_p(\mathbf{r}_p(t), t) + \mathbf{u}_s(\mathbf{r}_p(t), t) \tag{1}$$

$$\mathbf{r}_p(0) \quad = \quad \mathbf{r}_{0,p} \tag{2}$$

where $t$ denotes time and $\mathbf{r}_{0,p}$ in Equation (2) denotes initial position of $p$-th particle.
Terms of the right hand side of equation (1) are as follows. Term $\mathbf{u}_c(\mathbf{r}_p(t),t)$ denotes the Eulerian ocean current at particle
location $\mathbf{r}_p(t)$ at time $t$. In this study this term is obtained from the NEMO circulation model (Section 3.1.1) for forward prop-
agation simulations or from WERA HF radar observations for back-propagation simulations (Section 4.1). Term $\mathbf{l}_p(\mathbf{r}_p(t),t)$
denotes leeway of $p$-th particle at particle location $\mathbf{r}_p(t)$ at time $t$. Leeway term is computed from ALADIN winds (Section
3.1.2) as follows. Due to lift forces on the drifting object, its leeway is not oriented strictly downwind but has a crosswind
component as well or $\mathbf{l} = (l_\parallel, l_\perp)$, where $l_\parallel$ and $l_\perp$ are downwind and crosswind leeway components respectively. Experimen-
tal data however suggests an almost linear relationships between windspeed and downwind and crosswind leeway components
(Breivik and Allen, 2008). Therefore downwind leeway component can be parametrized as $l_\parallel = a_\parallel u_{10} + b_\parallel$, where $u_{10}$ denotes
windspeed. On the other hand the crosswind force can point both to the left or to the right of wind, depending on the orientation
and shape of the object in the wind field. Therefore crosswind leeway degenerates into left-drifting crosswind leeway compo-
nent $l_\perp^L = a_\perp^L u_{10} + b_\perp^L$ and a right-drifting crosswind leeway component $l_\perp^R = a_\perp^R u_{10} + b_\perp^R$. Coefficients $(a_\parallel, b_\parallel)$, $(a_\perp^L, b_\perp^L)$ and
$(a_\perp^R, b_\perp^R)$ are determined from observations as a least square linear fit between observed wind velocity and observed leeway
vector (Breivik and Allen, 2008; Allen and Plourde, 1999). The coefficients $(a_\perp^L, b_\perp^L)$ and $(a_\perp^R, b_\perp^R)$ are similar but not identical.
This linear regression also yields downwind, left-drift and right-drift standard deviations for each fit.
Term $\mathbf{u}_s(\mathbf{r}_p(t),t)$ on the right hand side of the equation (1) is the Stokes drift contribution, *i.e.* mean shift of a fluid particle
due to unclosed Lagrangian orbit of the particle in the gravity wave field. Note however that since coefficients $(a_\parallel, b_\parallel)$ and
$(a_\perp, b_\perp)$ are determined from observations, they already contain the Stokes drift contribution of the local wind sea in the ob-
served leeway. In our attempt to model object's leeway using downwind and crosswind leeway coefficients based on empirical
data from Allen and Plourde (1999), we have omitted the Stokes drift term from the initial value problem (1)-(2), as explained
in Breivik and Allen (2008).
OpenDrift is an open-source Python-based Lagrangian particle modelling code developed at the Norwegian Meteorological
Institute with contributions from the wider scientific community. Its Leeway() module implements leeway computation in the
fashion described in the previous paragraph, for further details see Breivik and Allen (2008) and Dagestad et al. (2018). Apart
from leeway computations OpenDrift supports a wide range of offline (*i.e.* with precomputed currents and winds) predictions
from oil spills and drifting objects to microplastics and fish larvae transport. Particle seeding is very convenient to use and its
Leeway module supports a wide range of object types with different lift and drag behaviour under current and wind forces
(Dagestad et al., 2018).
The object types used in this study were of two kinds that we believe are most adequate for leeway modeling in this particular
case. First drift object type was Person-in-water, corresponding to empirically determined (Allen and Plourde, 1999) downwind
slope of $a_\parallel = 1.93\%$, downwind standard deviation of $0.083\,\mathrm{ms}^{-1}$, right slope of $a_\perp^R = 0.51\%$, right standard deviation of $0.067$
$\mathrm{ms}^{-1}$, left slope of $a_\perp^L = -0.51\%$ and left standard deviation of $0.067\,\mathrm{ms}^{-1}$.
Second object type was Person-Powered-Vessel-2 (person with surf board), corresponding to empirically determined (Allen
and Plourde, 1999) downwind slope of $a_\parallel = 0.96\%$, downwind standard deviation of $0.12\,\mathrm{ms}^{-1}$, right slope of $a_\perp^R = 0.54\%$,
right standard deviation of $0.094\,\mathrm{ms}^{-1}$, left slope of $a_\perp^R = -0.54\%$ and left standard deviation of $0.067\,\mathrm{ms}^{-1}$.
The simulation was run in both cases for 48 hours using a second order Runge-Kutta scheme. Forcing data consisted of
NEMO currents and ALADIN SI 10m winds from the 00 UTC operational runs of both models, performed on 29 Oct 2019 at
ARSO.
At the time of the incident however, OpenDrift was not implemented at ARSO and could not be used. Due to the incident,
the pipeline of input data preparation and a specific drifting particle type OpenDrift computation was developed and is now
available to forecasters at ARSO as an internal web service. With ALADIN SI and NEMO fields (pre)computed operationally,
subsequent on-demand OpenDrift simulations take under ten minutes to complete.

## 203   4   Results and Discussion

### 204   4.1   Drift Trajectory Verification. Comparison of Survivors Trajectory to Backtracking Estimates from HF Radar

As noted above, survivor had no GPS or watch to keep track of his movements in space and time. Therefore his reconstruction
of the drift trajectory is burdened with error. What is known however is the exact location and time of his beaching: a beach in
Sistiana (Italy) on 30 Oct 2018 at 16 UTC. HF radar surface current measurements cover only the final part of the drift domain.
They can therefore, in themselves, not be used for a full forward-propagation simulation (starting at the accident location and
ending at the beaching location), but they can nevertheless be employed to perform Lagrangian back-propagation (upwind and
upstream advection backwards in time) during the final part of the drift.
Such simulation is of course limited to the HF system domain, described in section 2.2, but it should offer some insight
into the final part of the drift trajectory and serve as an independent check of survivor's trajectory estimate. To this end, HF
radar currents over the period of the drift were first gap-filled in space using nearest-neighbor interpolation and then gap-filled
in time using linear interpolation. Wind component for back-propagation was provided by the ALADIN atmospheric model
(see section 3.1.2) and remapped to HF radar grid in space and time. OpenDrift code was used to perform back-propagation
simulation and results are presented as particle numeric density per area, in similar fashion as in (Röhrs et al., 2018; Dugstad
et al., 2019). To ensure smooth maps of particle density, a large number (fifty thousand) of virtual particles of type "person
with surf board" were released from a circle within 500 m radius from the beaching location at beaching time. Particles were
then advected backward in time in 12-minute timesteps (0.2 hours) for 18 hours. Results of these simulations are depicted in
Figure 4, which shows particle density per density cell area. Density cells over which the particles are counted, were chosen
to be of 150 m × 150 m dimensions. This 10-fold reduction in cell computation area was done because the original HF radar
grid (1500 m × 1500 m, see Figure 3) is too coarse to produce smooth maps of particle density.
Figure 4 indicates two distinct pathways to Sistiana during the time of the drift: firstly we have the southern branch arriving
to the beaching site from the region south-southeast of the beaching site. Since we know that the survivor drifted into the Gulf
from the outside of the Gulf, and not from the region south-southeast of the beaching site, this pathway is not significant for
our analysis. Secondly, a northwestern branch of propagation is marked with a yellow line in Figure 4 and extends roughly
along the survivor's trajectory estimate. This pathway is qualitatively (spatially and temporaly) consistent with the survivor's
trajectory. Survivor's estimates of his location on 29 Oct 22 UTC and 30 Oct 10 UTC agree well with the computed virtual

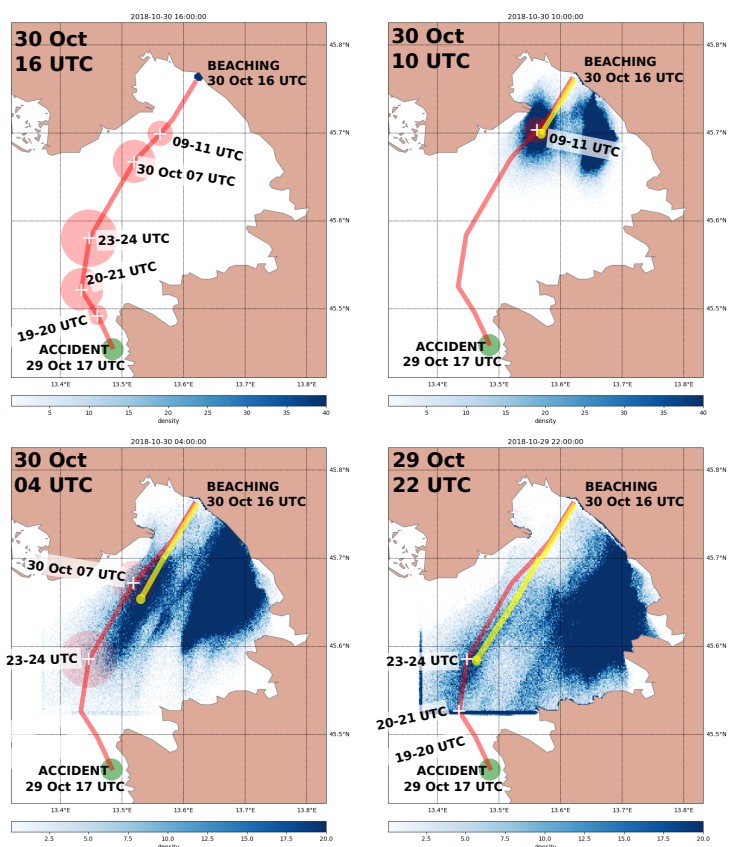

**Figure 4.** Temporal back-propagation of virtual particles from beaching location using HF radar measurements and ALADIN winds as inputs for OpenDrift model. Back-propagation starts at beaching location (top left panel). Particle spatial density is shown every six hours of the simulation, as denoted by timestamps in top left corner of each panel. Red line and superimposed dates are survivor's estimates of his trajectory (for clarity, only relevant timesteps of survivor's reconstruction are shown in each panel). Transparent red circles denote survivor's estimate of the error in his location at stated time. Yellow line is a reconstruction of survivor's trajectory from back-propagation simulations. Dark blue straight lines in the bottom right panel, appearing along the southwest corner of HF radar computational domain, result from accumulation of particles which cease to advect when they reach the outer limits of the HF radar domain.

particle density maps. During the night, when survivor reported feeling maximum distress, the back-propagation estimate of
trajectory is located a mile or two to the east of his reconstruction.
Additional back-propagation simulation was performed with NEMO modelled currents in identical fashion as described
above. This allows for a comparison with back-propagation due to the HF currents. This simulation is not shown here but is
depicted in Figure S1 in supplementary material. It indicates that NEMO currents tended to underestimate observed currents
throughout the drift. This underestimation is however most notable in the final hours of the drift, as commented also below in
Sections 4.2 and 4.3.

## 4.2 Marine Conditions from Observations and Models

In this section we present a qualitative analysis of marine conditions from available observations, and also marine drift results from both particle tracking models presented in Section 3.2.

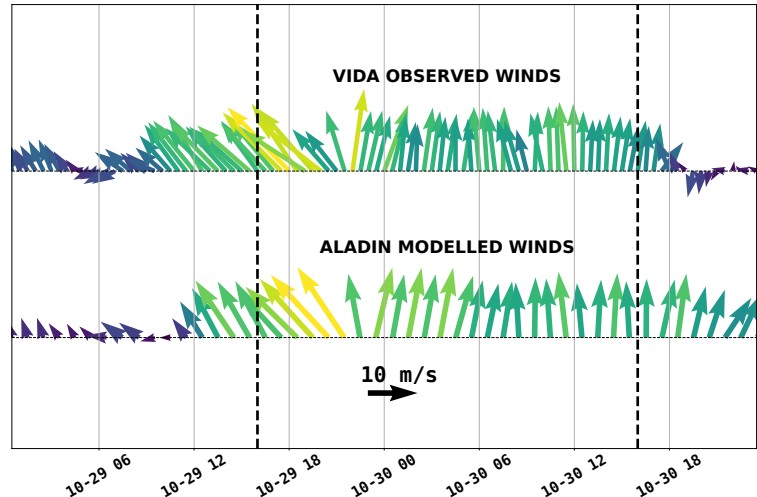

**Figure 5.** Arrow plots of observed and ALADIN SI modelled wind directions at Vida coastal buoy during 29 Oct 2018 event. Drift period is marked with dashed vertical lines. Arrows are colored by their wind speed.

Figure 5 depicts wind measurements and ALADIN SI modelled winds at the Vida coastal buoy (12 km northeast of the accident location, see Figure 1 b).) for the timewindow 29 - 31 Oct 2019. Qualitatively there is a very solid agreement between the two timeseries. Measured wind at Vida exhibits southeasterly 140° direction in the hours after the accident (left dashed line in Figure 5), followed by a shift to slight south-southwest 190° between 30 Oct 00 UTC and 04 UTC, and finally a southerly 180° direction during the day (all directions in the paper are stated in nautical notation, *i.e.* 0° marking north, 90° marking east). Wind speed is constantly around 15 - 20 $\mathrm{ms}^{-1}$.

HF observations in Figure 6 are presented as a qualitative check for the NEMO model surface currents during the 24 hours of the drift. HF measurements and modeled currents both exhibit eastward topographically constrained coastal current in the northern part of the Gulf between Grado and Soča/Isonzo rivermouth, with NEMO model tending to misrepresent and underestimate observations (as shown below however, wind drift was the main contribution to the drift). Absence of the coastal current on Oct 29th 22 UTC might be related to the model treatment of high Soča/Isonzo discharge, which in itself generates westward inertial current in that part of the modelling domain, and might be counteracting wind driven (eastward) currents. Verification of the NEMO model versus ten months of hourly HF radar currents (not shown in detail in this paper) yields a daily averages of bias in zonal velocity $< u_{NEMO} - u_{HF} >_{24h}$ between 0 and -2.5 $\mathrm{cms}^{-1}$ and a daily averaged bias for meridional velocity $< v_{NEMO} - v_{HF} >_{24h}$ between +2.5 and -2.5 $\mathrm{cms}^{-1}$. NEMO model underestimations during the limited period of this case study were unfortunately much larger: spatially averaged (over the HF domain) and temporaly averaged (over the

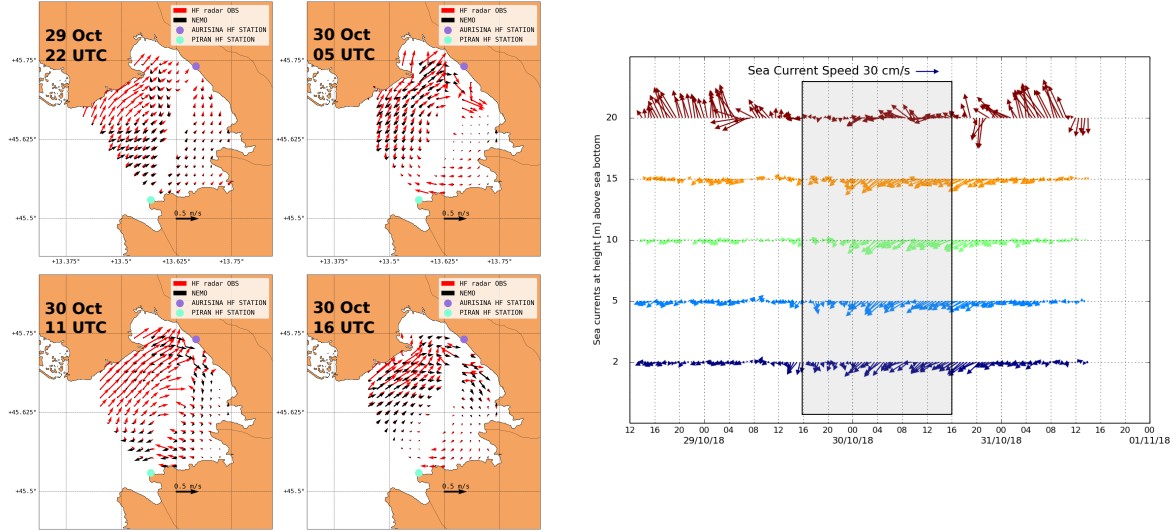

**Figure 6.** Left: HF radar measurements in the Gulf of Trieste during the period of the drift. Since there are gaps in surface current measurements, the closest observations to 29 Oct 22 UTC and 30 Oct 04, 10, 16 UTC are depicted. NEMO currents were bilinearly interpolated to WERA grid points. Arrow lengths from both fields are commonly scaled. Right: Arrow plot of ADCP measurements of ocean currents at Vida coastal buoy during 29 Oct 2018 event (shaded rectangle delimits the time window of the drift). Surface current timeseries is plotted in the top line.

period of the drift) NEMO biases amounted to -8.3 $\mathrm{cms}^{-1}$ for zonal velocity and a bias of -8.8 $\mathrm{cms}^{-1}$ for meridional velocity.
Focusing into the period of the drift, NEMO model performance was best in the beginning and worst at the end of the drift.
Over the first 18 hours of the drift, NEMO model exhibited $-8.2\,(8.1)\,\mathrm{cms}^{-1}$ bias in zonal (meridional) velocity, while during
the last 6 hours of the drift these biases amounted to $-11.4\,(-11.1)\,\mathrm{cms}^{-1}$. Additional forward-tracking simulations were also
performed to compare advection due to HF and NEMO currents. These simulations are not included here but are described and
depicted in Figure S2 in the supplementary material. They indicate that most of the error due to NEMO current underestimation
was accumulated in the final hours of the drift. This performance will have to be further addressed as a separate issue and needs
to be kept in mind when interpreting results below. On the other hand both the model and the HF measurements exhibit an
inflow over most of the surface area of the Gulf which indicates that the surface layer on Oct 29th 22 UTC was wind dominated,
see also Malačič et al. (2012).
Another common feature of NEMO currents and HF radar observations is the general anticyclonic character of the surface
circulation through the rest of the night and the following day. This is in contrast with the Northern Adriatic basin-scale
cyclonic current pattern during Scirocco episodes (not shown) and stems from the fact that Scirocco induced surface currents,
flowing north along the Istrian coast, typically branch upon hitting the northern end of the Adriatic basin. The eastward branch
of this wind driven current inflows into the Gulf of Trieste along the northern coastline. Such inflow, visible in modeled and
observed currents is therefore not unexpected during Scirocco episodes. As is further shown in the right panel of Figure 6, *in*

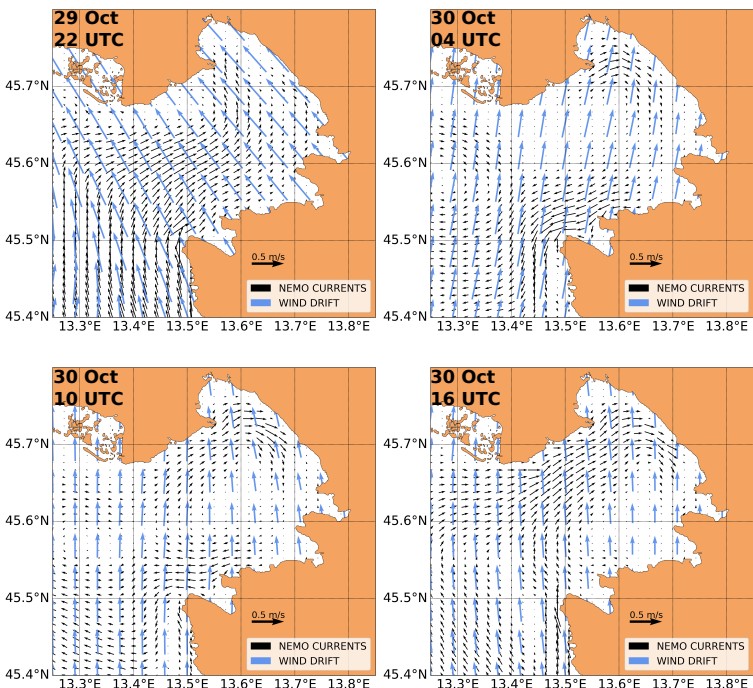

**Figure 7.** 6-hourly same-scale snapshots of NEMO currents (black arrows) and ALADIN SI 10 m wind $\mathbf{u}_{10}$ induced wind drift (blue arrows) over the period of the windsurfer's drift. Only purely downwind arrows with no crosswind departure from the ALADIN SI wind velocity direction are plotted, computed as $a_{\parallel}\mathbf{u}_{10}$ using OpenDrift "person in water" downwind slope $a_{\parallel} = 1.93$ %. Only every third wind point is plotted for clarity. Arrow lengths from both fields are commonly scaled and both arrow length units are m s$^{-1}$.

*situ* currents measured at Vida buoy also exhibit a westward direction over the entire water column during the timewindow of
the drift, and are therefore consistent with the overall anticyclonic character of the surface circulation, exhibited in the model
and radar surface current maps.
Figure 7 depicts current and wind drift inputs to both models over the period of the windsurfer's drift. The wind drift seems to
be the dominant driving factor of the windsurfer's drift, its speed being roughly double that of the surface currents. Wind drift
prior to (not shown) and at 22 UTC has a clear southeasterly direction (at Umag - offshore) at roughly 140-160°, consistent
with the windsurfer's experience and his inability to reach Savudrija in time. During the night the wind direction shifts into a
south-southwesterly to about 190°, also consistent with his experience. In the morning of 30 Oct 2018 and through the day,
the wind direction is predominantly southern at 180°. This is all in agreement with the direction shift measured at Vida buoy
(Figure 5).
NEMO currents at 22 UTC indicate northward direction along the coast of Istria and also a surface inflow along all but
the northernmost part of the opening of the Gulf of Trieste. The northernmost part along the northern coast of the Gulf most

likely shows no notable inflow due to inertial westward coastal current from the Soča/Isonzo river, which manifests itself as an outflow from the Gulf, confined to this part of the coast (see Figure 1 for the related river plume).

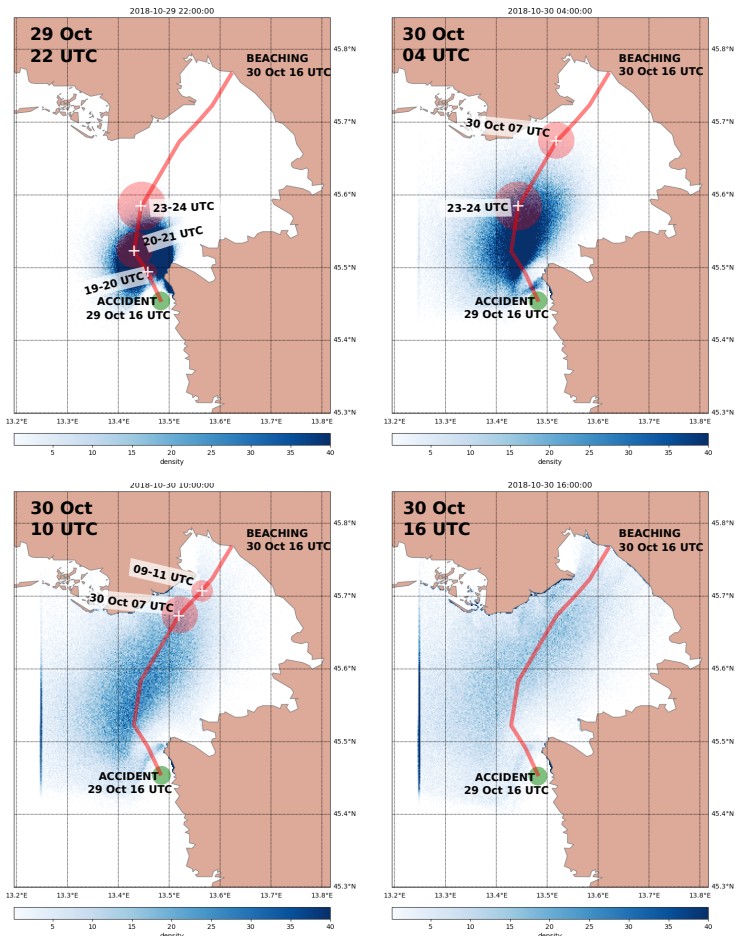

**Figure 8.** Lagrangian particle density [number of particles per cell area] from OpenDrift simulation of the "person in water" object type. Lagrangian simulation drift is depicted every 6 hours (indicated by a timestamp in the upper left corner of each panel) after the accident on 29 Oct 2019 16 UTC. Red line denotes drift trajectory as reconstructed by the survivor. White crosses and time inserts denote locations and times from survivor's trajectory estimate, while red circles around crosses denote survivor's uncertainty estimates of the respective location.

## 4.3 Lagrangian simulation results

In this section we present OpenDrift simulations with NEMO model current inputs and ALADIN SI 10m wind inputs during 19 Oct 2018 16 UTC and 30 Oct 2018 16 UTC. Simulations were performed running forward-propagation in time, starting particle drift from the accident location at Oct 2018 16 UTC.

OpenDrift results for drifting object type "person in water" are presented in Figure 8. Figure shows 6-hourly snapshots of
particle densities (number of particles per cell area), initially seeded in the green region at 29 Oct 2019 16 UTC. To ensure
smooth maps of particle density, a large number (fifty thousand) of virtual particles of type "person in water" were released at
the accident location at accident time and advected forward in time in 12-minute timesteps (0.2 hours) for 24 hours. Cells over
which the particles are counted were again chosen to be of $150 \text{ m} \times 150 \text{ m}$ dimensions. This reduction in cell computation
area was again done because the original NEMO grid resolution ($1000 \text{ m} \times 1000 \text{ m}$) is too coarse to produce smooth maps of
particle density. After 6 hours, at 22 UTC, the set of the particles envelops the estimated windsurfer location but the center of
gravity of the particle set is lagging southeast of survivor's estimated location.
Shift in the wind direction from southeast to south-southwest (see Figure 5), occuring sometime after 29 Oct 22 UTC and
lasting until 04 UTC, causes a corresponding shift in particles' drifting directions and a stretched dispersal of the particle set
along the survivor's trajectory estimate.
First particles are beaching on the northern shore of the Gulf between 04 UTC and 10 UTC. This predominantly occurs
between Grado and the Soča/Isonzo river mouth. Particles in the Gulf are propagating along the reconstructed trajectory, but
with increasing lateral and axial extent. At 10 UTC the set is dispersed over northwestern half of the Gulf of Trieste and are
stretched roughly along the survivor's trajectory. While majority of particles lag behind the survivor's estimated location the
set does extend over survivor's estimated location which is enveloped by the forefront of the particles set.
After 24 hours the particles set is almost homogeneously dispersed over the northwestern half of the Gulf, with some of the
particles beaching within 2 km of the actual beaching location.
OpenDrift results for drifting object type "person with surf board" are presented in Figure 9. After 6 hours, at 22 UTC, the set
of the particles envelops the estimated windsurfer location and the center of gravity of the particle set is closer to the survivor's
estimated location than in the "person in water" case. This particle set is also overlapping with the higher density region of the
northernwestern pathway from HF radar currents back-propagation simulation result at 29 Oct 2018 22 UTC presented in the
bottom right panel of Figure 4.
Shift in the wind direction from southeast to south-southwest (see Figure 5), occuring sometime after 29 Oct 22 UTC and
lasting until 04 UTC, again causes a corresponding shift in particles' drifting directions but the dispersal of the particle set along
the survivor's trajectory estimate is somewhat lesser than in the "person in water" case. At 04 UTC the majority of the particles
is lagging behind (*i.e.* is mostly located southwest of) both survivor's location estimate and also behind the densest region from
the northwestern branch of the back-propagation simulation (bottom right panel in Figure 4). This is consistent with the fact
that NEMO modelled currents are underestimating HF radar measurements used for back-propagation simulations.
At 10 UTC the particle set is dispersed between Grado and Soča/Isonzo rivermouth, again lagging behind both survivor's
location estimate and the northwestern branch of the back-propagation simulation (top right panel in Figure 4). When compared
to "person in water" scenario, this particle set is however more clearly localized along the northern shore of the Gulf.
After 24 hours the particles set is densest around the Soča/Isonzo rivermouth, but with a clearly visible streak of particles
beaching within 2 km of the actual beaching location. This higher localization represents some improvement over the entire

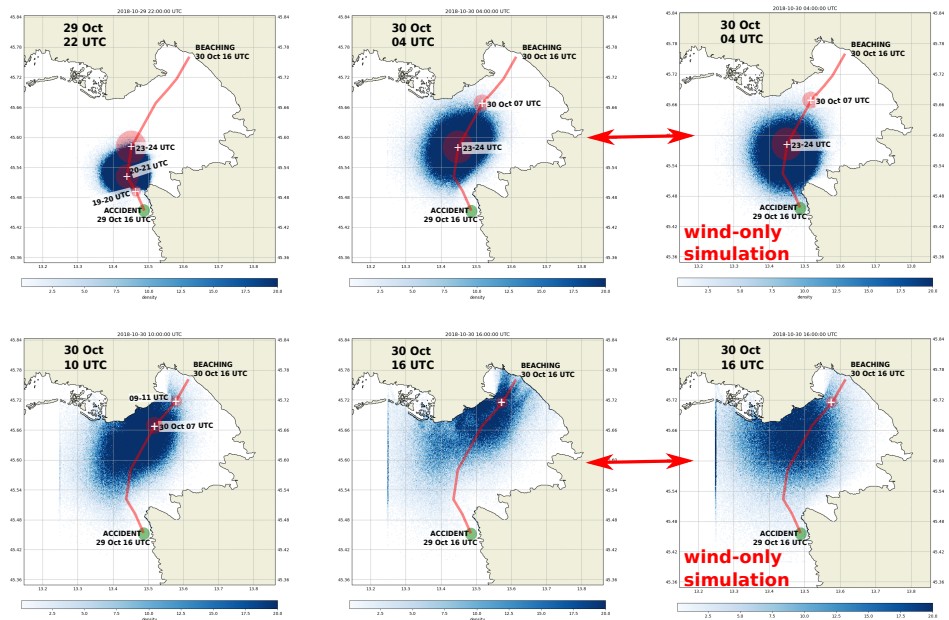

**Figure 9.** Same as Figure 8 but for "person with surf board" object type. Rightmost column depicts simulation results with wind-only input (and ocean currents set to zero) after 12 hours and 24 hours of the drift.

northwestern half of the Gulf of Trieste, indicated by "person in water" simulation. In any case a quantitative comparison is performed below to further elucidate performances of both drift simulations.

Figure 9 contains also a third separate column which depicts results of a wind-only simulation (with ocean currents artifically set to zero) after 12 hours and 24 hours of drift, *i.e.* at 30 Oct 04 UTC and 16 UTC respectively. Comparison with the first two columns (depicting full simulations with both winds and currents) demonstrates ocean current influence to particle dispersal. Under rather homogeneous wind conditions (see Figure 7) particle dispersal due to wind is highly isotropic throughout the simulation period. Being highly inhomogeneous itself, ocean current pattern in the Gulf adds asymmetry to the wind dispersal pattern. This effect elongates somewhat the slick of particles and advects them further along the Italian coast closer towards Sistiana.

Given available data (or lack thereof) a quantitative comparison between the drift simulations can only be based on the beaching point, which is known. To pursuit this we calculated the distribution of stranded and active (non stranded, still in the water column) particles and plotted its histogram over distances from the beaching location at beaching time and also 6, 12, 18 and 24 hours after the beaching time. Distributions at beaching time and particle accumulation after the beaching time are depicted in Figure 10.

Panel A2 in Figure 10 indicates that at beaching time the distribution maximum of active "person with surf board" particles is positioned about 12 km from the beaching site. It is also positioned 15 km closer to the beaching point than the distribution

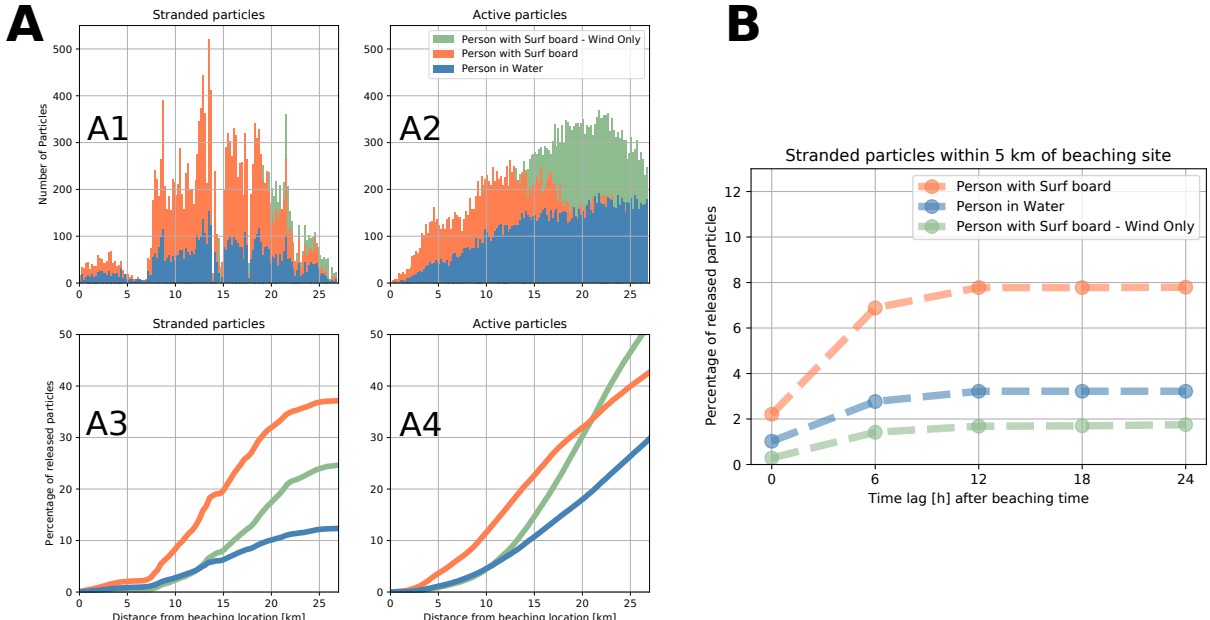

**Figure 10.** Panels A1 and A2: Stranded (A1) and active (A2) particle distribution over distances from the beaching location at beaching time. Panel A3(A4): cummulative particle distributions of stranded (active) particle distribution, shown in A1 (A2), over distances from the beaching location at beaching time. Panel B: time dependence of acummulated percentage of particles within 5 km of the beaching site.

maximum of "person in water" particles. This indicates a) better performance of "person with surf board" particles, and b) a
time lag in the movement of all types of particles. As mentioned above, this is very likely due to the NEMO model surface
current underestimation during the event - this claim is further backed by the fact that WERA HF back-propagation simulations
in section 4.1 seem temporaly consistent with survivor's estimate (Figure 4) and show little lag after 18 hours of the drift.
These conclusions are also implied by the particle distributions in Figures 8 and 9: at beaching time, "person in water"
particles are dispersed over a much wider area than those of "person with surf board" type. Panel A2 of Figure 10 reflects that.
However, and regardless of this lag, when focusing on the accumulation of stranded particles (panels A3 and B of Figure
10), we see that at beaching time about twice as many "person with surf board" type particles stranded within 5 km of the
beaching point than those of "person in water" type. The same holds for particles stranding within 10 km radius. Within 20 km
radius this ratio triples. These results quantitatively substantiate claims of better performance of the "person with surf board"
particle type for this case study.
Panel B on Figure 10 shows percentage of stranded particle within 5 km distance of the beaching site in the hours after the
beaching. This percentage saturates on the scale 6 hours, giving us an estimate for the time lag between the actual survivor
beaching and beaching of the majority of simulation particles.
We conclude this section with a brief comment on wind-only simulations with "person with surf board" type. These sim-
ulations under homogeneous wind conditions exhibit highly isotropic spatial dispersion of particles, unlike the two scenarios

which take into account ocean currents. This leads to slower accumulation of particles within 5 km radius of the beaching point (panels A3 and B of Figure 10). At these distances and by this metric, wind-only simulations are the worst performer of all three. Without putting too much weight on wind-only simulation - this does indicate the importance of topographically constrained ocean currents in semi-enclosed basins like the Gulf of Trieste even in seemingly wind-dominated situations.

## 5  Conclusions

In the paper we present a modeling analysis of the 24-hour marine drift by the windsurfer whose mast broke on 29 Oct 2018 16 UTC, during a 29 Oct 2018 Scirocco storm in the Northern Adriatic. We conduct an interview with the survivor in order to reconstruct his trajectory and its uncertainty. The survivor knows the coast of the Gulf of Trieste very well, but had no GPS or watch on him during the drift. His reconstruction of the drift trajectory is therefore burdened with error. To estimate this error we used HF radar surface current measurements, which cover the second half of his drift, and employed them for upwind and upstream temporal back-propagation simulations starting at the beaching site at beaching time, both of which are exactly known. These back-propagation simulations were found to be largely consistent with survivor's reconstruction, offering some confidence that while not perfect, the reconstructed trajectory can nevertheless serve as a qualitative guide for Lagrangian tracking.

We then present ocean circulation (NEMO), atmosphere (ALADIN SI) and OpenDrift Lagrangian tracking models, used to perform forward-propagation simulations of this trajectory, starting from the accident location. We present available marine measurements (regional coastal buoy Vida and HF surface current radar) to qualitatively assess marine conditions in the Gulf of Trieste during the period of the drift.

OpenDrift Lagrangian tracking model was employed using two types of marine drift parametrizations: "person in water" and "person with surf board". Stokes drift from a wave model was not explicitly included in OpenDrift forcing data since these effects are already implicitely present in the downwind/crosswind drift parametrizations, deduced from observations (Breivik and Allen, 2008).

To quantify performance of both drift parametrization types, we calculated distributions of particle distances from the beaching location for both object types. Simulations using object type "person with surf board" yield best performance, with highest number of particles stranded within 5 km of the beaching location. Distribution maximum of "person with surf board" particle type is positioned about 15 km closer to the beaching point than the distribution maximum of "person in water" particle type. Both scenarios however lag behind the estimated drift which most likely results from NEMO model underestimation of surface currents during the event. For both drift parametrization types accumulation of particles, stranded within 5 km of the beaching location, saturates roughly six hours after the actual beaching time.

A control run of wind-only forcing was also simulated and this setup was the worst performer of all three, indicating the importance of topographically constrained ocean currents in semi-enclosed basins like the Gulf of Trieste even in wind-dominated situations.

Results in these paper indicate that any rescue response in the 29 Oct 2018 case would certainly benefit from OpenDrift
simulations using "person with surf board" object type. However, while one can clearly benefit from using the most appropriate
drift parametrization, lack of information during an actual event often complicates the decision on which parametrization to
use.
It is also worth mentioning that given the location of the accident, a drift under Bora wind conditions seems substantially
more dangerous. Bora is typically much colder and can, regardless of its short fetch, generate comparable marine conditions in
Northern Adriatic, but its nautical direction is $60°$, *i.e.* completely offshore in Northern Istria. Marine drift initiated in Umag
(or, more likely, at a popular Bora windsurfing spot near the Cape of Savudrija) during the Bora would have lasted days, and
possibly more than a week if the person would get advected westward far enough to join Western Adriatic Current flowing
southward along the Italian coast. Reliable and operational circulation models, coupled to calibrated Lagrangian tools like
OpenDrift, would be an invaluable decision support for any rapid rescue attempt.
*Author contributions.* A.F. and M.L. set up NEMO for the Adriatic basin. A.F. performed NEMO simulations. S.E. performed NEMO
verification against HF radar data. M.L., S.E. and A.F. performed the OpenDrift simulations. S.E., C.R.S., D. D. and M. L. analyzed the HF
radar data. M.L. devised the work plan and wrote the paper. All authors contributed to the final editing of the paper.
*Competing interests.* Authors declare no competing interests.
*Acknowledgements.* The authors would first and foremost like to thank the survivor of the incident, Mr. Goran Jablanov, for his willingness
to respond to the interview request and to reconstruct the trajectory of his drift as accurately as possible. The authors would like to thank
Sašo Petan (ARSO) for providing Soča/Isonzo river runoff at Solkan station. The authors would like to thank all technicians and engineers
at our institutions for enabling and supporting the research work. M. L. would like to thank Augusto Sepp Neves for useful discussions.
M. L. wishes to acknowledge financial support of the Slovenian Research Agency project grant J1-9157: "Drivers that structure coastal
marine microbiome with emphasis on pathogens – an integrated approach". The paper benefited greatly from comments by two anonymous
reviewers.

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
