# Peer review of "Lagrangian Modelling of a Person lost at Sea during Adriatic Scirocco Storm of 29 October 2018"

_Natural Hazards and Earth System Sciences, 2019_

## Referee Comment (RC1) · Anonymous Referee #1 · 30 Jan 2020

Review of : Lagrangian trajectory modelling for a person lost at sea. . ..

By Licer et al

The subject of the paper is relevant, since it evaluates the performance of two Lagrangian models versus a real accident with a person lost at sea during an extreme meteorological event. The paper implementation though lacks in my opinion of several crucial elements. I therefore recommend re-submission, and ask the authors to address the points detailed in the following.

1) In Section 1, the authors introduce the reconstructed trajectory of the survivor (Fig.1b), that they use to test the two proposed models. I find the trajectory and the

associated uncertainty (500 m radius) quite arbitrary. Was the survivor in possession of a GPS, or at least of a watch (time information)? If so, this should be mentioned, and if not, how did he estimate time and position of the trajectory?

In absence of solid information, I think that the evaluation should be based mostly on the only secure data point, which is the beaching point (in space and time) of the survivor, rather than trying to quantitatively match the specific positions evaluated along the trajectory.

2) The description of the two Lagrangian models is inconsistent, the conceptual differences between the two are not clearly outlined, and a number of basic information that are then used later on in the paper, are not provided. Specifically, why is the particle equation written for Flow Track ( Section 4.2, eq 1) and not for OpenDrift (Section 4.1)? If the basic equation is the same, it should be introduced in Section 4.1 and the differences in parameterizations and considered processes should be discussed up front, for instance in terms of wind drag and lift, Stokes, turbulence etc..

Also the model implementation should be better discussed. How many particles are typically launched in each model? (this is mentioned for Flow Track but not for Open-Drift) How are the results diagnosed? In Section 5, it is mentioned that for OpenDrift a Rescue Area (RA) is computed as a polygon based on particle location, while in Section 5 and 6, it is mentioned that a RA cannot be computed for Flow Track... I do not understand why is it so, and I think it should be better explained. Also, all these aspects should be presented up front in Section 4, rather than at the end of the paper.

3) The discussion of model evaluation in Section 5 is in my opinion very unclear. The paragraphs at pg12 and 13 for PIW and PPV are simply repeated with the RA values changed...What are the grey shading areas shown in Fig. 8-10 and how are they computed? I do not see a clear difference in the three cases. The authors seem to favor the results of OpenDrift PPV because the RA is more reduced (even though it still covers the whole Gulf?), but the RAs are not shown in the figures.
[Figure]

In general, as mentioned in point 1) I think that the quantitative assessment should be based mostly on the beaching point. What is the distribution of beached particles in time and space for the 3 configurations?

4) The conclusions (Section 6) are in my opinion not satisfactory. The authors mention that OpenDrift is more suitable for Search and Rescue applications because it is more operational, i.e. it has a classification for object parametrizations, and it provides RAs. But these points were known from the beginning, given the model characteristics! What is the added value of the comparison?

I think the authors should discuss in an objective way the results, and indicate strength and weakness of each model with respect to the actual performance. Of course, it is also important to point out the shortcoming of Flow Track in terms of operational performance, but indeed that was a given and we did not need this exercise to reach this conclusion. . .

5) Finally, I would like to make a general comment. From the patterns of currents and wind (Fig.5,7), it looks like the trajectory of the survivor was likely to be strongly influenced by the ocean currents (that facilitate the entrance inside the Gulf,) while the trajectories of both models tend to overestimate the wind influence (that moves them more to the north-west).

Indeed Fig.6 shows that the wind input in the Lagrangian models is approximately double with respect to the currents (please clarify the dimensions of the variables in that Figure: are they velocities or are they model inputs somehow normalized?). On the other hand Fig.5 shows that the NEMO current amplitude is underestimated with respect to the HF radar.

So, it is possible that improving NEMO results would greatly improve the trajectories of the Lagrangian models. Alternatively, could the HF radar results be used as inputs in the Lagrangian models? I understand that there is a permanent gap in the middle of the Gulf, likely due to the GDOP, but probably gap filling techniques can be used to

ensure a more extended coverage. The authors should explore these aspects.

---

## Referee Comment (RC2) · Anonymous Referee #2 · 8 Feb 2020

General Comments

In this work the potential trajectory followed by a windsurfer lost at sea in the surroundings of the Gulf of Trieste during a storm that took place in October 2018 is simulated through a Lagrangian approach.

To this end, high resolution oceanic and atmospheric models are used as input for two Lagrangian tools: OpenDrift and FlowTrack. Despite using the same numerical integration scheme (Runge-Kutta 2nd order) they present some differences. Likely the most important is that Open Drift offers pre-calibrated coefficients for downwind and crosswind components more suitable to simulate the drift of a person at sea, specially
if the wind drift is dominant. As a result, the final distribution of particles simulated by OpenDrift match better the (tentative) inferred trajectory reconstructed by the authors from an interview with the windsurfer.

My impression is that this paper is based on a compelling idea and that results can be potentially useful at regional and operational level. English is good and one key strength is that model data (especially ocean currents) have very high resolution. However, I feel that this version of the manuscript needs some reorganization, a more accurate description of methods, a more elaborated assessment of the different contributions of wind/ocean currents/(waves?) terms to the simulated trajectories, and an improved treatment of uncertainties and search and rescue areas, before being considered suitable for publication.

In the below lines I elaborate further my above overall impression.

Major Comments

A- It is rather weird (and a bit confusing) to describe wave data from ECWAM model when it is not used later in the Lagrangian simulations. Indeed I think it is more logic to start in Section 2 with the quite general equations of Lagrangian particle tracking (current Section 4), and later describe in Section 3 observations and model data. In this way the fact that you neglect the Stokes drift is clearly stated and there is no need to include a description of the wave model in Section 3. Then in Section 4 you could show the results on the validation of model data with observations, and so on.

B- You say that you do not consider the Stokes drift for your simulation because wavelengths are significantly larger than the size of a person/windsurf table. However, as far as I understand in the case of microplastics, wavelengths are proportionally even larger but the Stokes drift has an important role on their distribution (e.g. van den Bremer and Breivik; Onink et al., 2019). Am I wrong? I think that it is likely simpler to say that the Stokes drift is generally a second (or third) order term in the Adriatic Sea in terms of magnitude (e.g. see Fig. 1d within Onink et al., 2019) . Indeed it would be a good

exercise to verify this point with an in-situ wave buoy or with your wave model data.

C- It is not explained which expression you use to simulate turbulent effects (random numbers) at each time step. Is it based on a uniform or a normal distribution? Is it different for each direction? Is the maximum number of 0.02 m/s based on observations or a model assessment? Indeed in coastal regions where strong gradients in ocean currents exist the magnitude of diffusivity can change significantly between relatively close grid cells depending on the ocean features. Please clarify this point.

D- With respect to the validation of the model data with observations, why do not show a comparison between the ADCP data (Figure 6, violet arrows) and the closest grid cell ocean model velocity? This would help to provide some numbers on the discussed underestimation of modeled currents. Also, do you have any reference in which HF radar velocities have been validated? Add it (them) to Section 2.2.

E- Other unclear points are:

- How do you estimate the light red circles in Fig. 8-10?

- Why particles are initially deployed in a rectangular shape?

- You deploy initially 480 particles, one for each value of Lp(theta), however in this way the uncertainty introduced by their different initial location is not assessed. What is the impact of the uncertainty in the initial position on the final search and rescue area for each Lp(theta)? Is it significant?

F – I find that would be interesting to show also the distribution of simulated particles with only wind drift/only ocean currents, and to estimate how large is the dispersion of particles (final area of search and rescue) for both cases. It would show graphically how predominant is the wind drift in the advection of particles and the lag with only ocean currents.

G- My major criticism to this work is the approach you follow to show your results. Considering that it is aimed to be useful for search and rescue (SAR) tasks and, therefore,

time is critical, I think that to show trajectories is far less useful than to show areas (or contours) of accumulated probability constructed with suitable mathematical functions. In terms of accumulated probability the areas for search and rescue tasks can be more easily prioritized. Your estimated areas after just few hours of simulation look pretty large to be useful for search and rescue tasks. Additionally, the probabilistic approach naturally includes the fact that there are uncertainties everywhere. Even more, probability contours can have a bimodal distribution, while your polygon seems to include all particles inside irrespective of the spatial holes among them. For example, this approach can be found in Abascal et al., (2010).

You need to convince me that for SAR tasks your current approach is reasonable enough, otherwise I suggest to redo your Fig. 8-10 with a probabilistic perspective (showing e.g. contours of 50%/70%/90% of accumulated probability estimated from the distribution of your particles), which is relatively easy to implement. In the latter case I suggest to remove the word "trajectory" from the title.

Other Comments

Title. Change "for" by "of".

Ln 2. Suggest to change "He was drifting" by "He drifted".

Ln 6. We "modeled".

Ln 52. Unclear how you estimate the +/- 500 m of error.

Ln 65. "By the time he is entering" . . .

Ln. 218. Remove point after "day"

Ln. 218. "all directions..."

Ln. 224. "generates a westward initial current".

References

van den Bremer, T. S. and Breivik, Ø. Stokes drift (2018). Philosophical Transactions of the Royal Society A: Mathematical, Physical and Engineering Sciences, 376, 2111, 10.1098/rsta.2017.0104,

Onink, V., Wichmann, D., Delandmeter, P., van Sebille, E. ( 2019). The role of Ekman currents, geostrophy, and stokes drift in the accumulation of floating microplastic. Journal of Geophysical Research: Oceans, 124, 1474– 1490. https://doi.org/10.1029/2018JC014547

Abascal, A., Castanedo, S., Medina, R., Liste, M. (2010). Analysis of the reliability of a statistical oil spill response model. Marine Pollution Bulletin 60 (11), 2099–2110.

---

## Author Comment (AC3) · 2 Mar 2020

The comment was uploaded in the form of a supplement:
https://www.nat-hazards-earth-syst-sci-discuss.net/nhess-2019-362/nhess-2019-362-AC3-supplement.pdf

---

## Author Response (AR1)

Authors' response to the Editor and both Referees
regarding
Review of : Lagrangian trajectory modelling for a person lost at sea. . ..
By Licer et al

The authors would first like to thank both the editor and both reviewers for taking the time to read and comment on the paper.  We believe their comments led to a much improved and clearer revised version of the paper. Before we address their respective comments point-by-point below, we would like to briefly recapitulate major modifications to the paper content here:

- Independent verification of survivor's trajectory estimate is performed, using WERA HF radar back-propagation simulations from the beaching site. Both trajectories were found to be qualitatively consistent.
- A 10-month verification of the NEMO model was performed against HF radar data in the Gulf of Trieste. This allows us to put numbers on NEMO performance during the drift. Unfortunately this performance was below-average and this is now reflected in the paper where meridional and zonal model biases are stated.
- NEMO model description has been revised to indicate that HF radar data assimilation is not implemented at the time
- FlowTrack sections of the paper were left-out of the revised version since we agree with the review that OpenDrift-FlowTrack comparisons lacked added value to merit inclusion.
- Current-only and wind-only simulations were performed with "person with surf board" drifter type. Wind-only simulations are now included in the revised version of the paper and analyzed alongside full wind+current simulations.
- Quantitative estimation of performance of each simulation was added to the paper. We base these estimations on stranded and active particle distributions over distances from the beaching site. This allows us to quantify somewhat the performances of each simulation type even in the context of poor NEMO model performance. "Person with surf board" scenario yields also quantitatively the best results.
- Time lag of particle stranding at the beaching point was assessed from temporal dependence of particle distributions near the beaching site.

Our point-by-point responses are below. Reviewer suggestions are typed in **bold-face**, author responses (AR) are normally typed.

Editor

Review of : Lagrangian trajectory modelling for a person lost at sea. . ..
By Licer et al

To me your recommendations in terms of effectiveness of the search strategy and for the survival of the windsurfer are not clear, neither in the abstract, nor in the conclusion. You simply comment that "search and rescue response should be as rapid as possible". This seems a rather trivial conclusion.

We thank the editor for this comment. We agree. This dictum has now been left out of the paper. Based also on other reviews, the whole paper was thoroughly rewritten to include less ambiguous and more quantitative analysis of Lagrangian model performance.

An example of questions that come to my mind is whether the search area (linearly increasing with time) could be effectively searched by a rescue vessel with reasonable chance to spot the windsurfer. Further, was the time needed for completing the simulation compatible with a timely rescue of the person lost at sea?

Thank you for this comment. We have now omitted the search-and-area formulations from the revised manuscript. We are not experts on search and rescue strategies, but we discussed this question with the Slovenian Civil Rescue service: first 15 hours after the incident, the rescue attempt was not possible due to severe conditions. They did in fact attempt marine and aircraft search and rescue action but it was not successful. Nevertheless, and given the limited dimension of the Gulf of Trieste, it was their opinion that timely rescue would have been possible – especially when guided by a reliable Lagrangian simulation.

It is not immediately clear from your text, but it seems that the whole model chain works in analysis mode, i.e. assimilating observations. Please, clarify this point in the text. If the results that you use are the results of an analysis, this would not be practically available during the actual search. How much does the result deteriorate if forecast fields are used?

We thank the editor for this comment. The text has been amended to show that NEMO does not at present have any data assimilation. Verification of the NEMO model versus ten months of hourly HF radar currents (not shown in this paper) yields a bias in zonal velocity between 0 and -2.5 cm/s and a bias between +2.5 and -2.5 cm/s for meridional velocity. NEMO model underestimations during the limited period of this case study were however even larger: spatially averaged (over the HF domain) and temporaly averaged (over the period of the drift) NEMO biases amounted to -6.3 cm/s for zonal velocity and a bias of -9.2 cm/s for meridional velocity. NEMO setup therefore exhibited below-average performance during the period of interest. This is now included in the paper. It also indicates that the model would benefit from HF radar data assimilation and we will be focusing on this in the future.

Anonymous Referee #1

Review of : Lagrangian trajectory modelling for a person lost at sea. . ..
By Licer et al

The subject of the paper is relevant, since it evaluates the performance of two Lagrangian models versus a real accident with a person lost at sea during an extreme meteorological event. The paper implementation though lacks in my opinion of several crucial elements. I therefore recommend re-submission, and ask the authors to address the points detailed in the following.

AR: We thank the reviewer sincerely for taking the time to read and comment on the paper!

1) In Section 1, the authors introduce the reconstructed trajectory of the survivor (Fig.1b), that they use to test the two proposed models. I find the trajectory and the associated uncertainty (500 m radius) quite arbitrary. Was the survivor in possession of a GPS, or at least of a watch (time information)? If so, this should be mentioned, and if not, how did he estimate time and position of the trajectory? In absence of solid information, I think that the evaluation should be based mostly on the only secure data point, which is the beaching point (in space and time) of the survivor, rather than trying to quantitatively match the specific positions evaluated along the trajectory.

AR: We thank the reviewer for this important question – we followed reviewer's instructions closely. No, the survivor was not in posession of a GPS or a watch. Therefore the reviewer is right to point out the arbitrary nature of his estimates. This reviewer's comment has  - in conjunction with demands from the second reviewer – led to our independently verification of the survivor's drift trajectory. As described in a new passage in the paper this was done by interpolating in space and time the HF radar currents during the period of the drift, and to then use these measurements to compute Lagrangian *back*-propagation from the *beaching* location as the reviewer suggested. This simulation yields a trajectory which seems largely consistent with survivors estimate. We have explained trajectory verification in a separate section and plotted the results in Figure 4. We hope the reviewer will agree that this substantiation gives  some credibility to the survivor's estimate and that we can use it as a qualitative orientation also during the drift, not only at the beaching point.

2) The description of the two Lagrangian models is inconsistent, the conceptual differences between the two are not clearly outlined, and a number of basic information that are then used later on in the paper, are not provided. Specifically, why is the particle equation written for Flow Track ( Section 4.2, eq 1) and not for OpenDrift (Section 4.1)? If the basic equation is the same, it should be introduced in Section 4.1 and the differences in parameterizations and considered processes should be discussed up front, for instance in terms of wind drag and lift, Stokes, turbulence etc..
Also the model implementation should be better discussed. How many particles are typically launched in each model? (this is mentioned for Flow Track but not for OpenDrift) How are the results diagnosed? In Section 5, it is mentioned that for OpenDrift a Rescue Area (RA) is computed as a polygon based on particle location, while in Section 5 and 6, it is mentioned that a RA cannot be computed for Flow Track. . . I do not understand why is it so, and I think it should be better explained. Also, all these aspects should be presented up front in Section 4, rather than at the end of the paper.

AR: We thank the reviewer for this comment. We agree that the two Lagrangian models are different in many respects and this is often an issue when comparing different model performances. After both reviewers pointed out this comparison as a weakness of the paper, we decided to leave FlowTrack section out and instead deepen the OpenDrift part.

Now there is only one Lagrangian model in the paper (OpenDrift), but it is presented in more depth and used in more ways. It is used for back-propagation trajectory verification on HF currents and subsequently for forward-propagation with modelled currents. We hope this lessens the confusion and clarifies the paper.

However, to answer the reviewer's question about RA in FlowTrack: the RA can of course be computed also for FlowTrack which, as reviewer mentions, should indeed have been pointed out in the first version of the paper. However the particles in Flowtrack simulations form a very thin sickle(y) geographic shape so the actual RA is misleadingly small and therefore not very telling. This question however has now been put aside since FlowTrack was omitted in the new version of the paper.

3) The discussion of model evaluation in Section 5 is in my opinion very unclear. The paragraphs at pg12 and 13 for PIW and PPV are simply repeated with the RA values changed. . .What are the grey shading areas shown in Fig. 8-10 and how are they computed? I do not see a clear difference in the three cases. The authors seem to favor the results of OpenDrift PPV because the RA is more reduced (even though it still covers the whole Gulf?), but the RAs are not shown in the figures. In general, as mentioned in point 1) I think that the quantitative assessment should be based mostly on the beaching point. What is the distribution of beached particles time and space for the 3 configurations?

AR: We thank the reviewer for the suggestion. We have followed it and we now include the calculation of the distribution of stranded and active (still in the water column) particles and plotted its histogram over distances from the beaching location at beaching time for both drifter types. These plots are now depicted in Figure 10 in the paper. The following analysis was added to the paper:

"Top right panel in Figure 10 indicates that the distribution maximum of "person with surf board" drifters is positioned about 12 km closer to the beaching point than the distribution maximum of "person in water" drifters. The fact however that the distribution maximum of "person with surf board" drifters is also positioned about 12 km from the beaching point indicates that there is a lag in the movement of these drifters. As mentioned above, this is very likely due to the NEMO model surface current estimation during the event. These conclusions are also consistent with spatial particle distributions in Figures 8 and 9: at beaching time, "person in water" particles are dispersed over a much wider area than those of "person with surf board" type. Top right panel of Figure 10 reflects that.

However, and regardless of this lag, when focusing on the accumulation of stranded particles (bottom left panel of Figure 10), we may reach a conclusion that at beaching time about twice as many "person with surf board" drifters stranded within 5 km of the beaching point than those of "person in water" type. The same holds for particles stranding within 10 km radius. Within 20 km radius this ratio triples. All these results quantitatively substantiate earlier qualitative claims of better performance of the "person with surf board" drifter type for this case study."

Included in the mentioned section are also wind-only simulations results, treated in the same fashion as indicated in the passage above.

We also performed new simulation runs beyond the beaching time to estimate time-lag of particle arrivals. For all drifter types, stranded particle distributions begin to saturate roughly six hours after the beaching time. This result has now been included as the B panel in Figure 10.

4) The conclusions (Section 6) are in my opinion not satisfactory. The authors mention that OpenDrift is more suitable for Search and Rescue applications because it is more operational, i.e. it has a classification for object parametrizations, and it provides Ras. But these points were known from the beginning, given the model characteristics! What is the added value of the comparison?
I think the authors should discuss in an objective way the results, and indicate strength and weakness of each model with respect to the actual performance. Of course,is also important to point out the shortcoming of Flow Track in terms of operational performance, but indeed that was a given and we did not need this exercise to reach this conclusion. . .

AR: We agree that there was too little added value to the original model comparison and that it does not warrant publication. Flowtrack is now out of the paper.

5) Finally, I would like to make a general comment. From the patterns of currents and wind (Fig.5,7), it looks like the trajectory of the survivor was likely to be strongly influenced by the ocean currents (that facilitate the entrance inside the Gulf,) while the trajectories of both models tend to overestimate the wind influence (that moves them more to the north-west).
Indeed Fig.6 shows that the wind input in the Lagrangian models is approximately double with respect to the currents (please clarify the dimensions of the variables that Figure: are they velocities or are they model inputs somehow normalized?).
On the other hand Fig.5 shows that the NEMO current amplitude is underestimated with respect to the HF radar. So, it is possible that improving NEMO results would greatly improve the trajectories of the Lagrangian models. Alternatively, could the HF radar results be used as inputs in the Lagrangian models? I understand that there is a permanent gap in the middle of the Gulf, likely due to the GDOP, but probably gap filling techniques can be used in it in to ensure a more extended coverage. The authors should explore these aspects.

AR: We agree with this comment and thank the reviewer for it. This comment, in conjunction with the first comment, led to gap-filling of the HF radar data and to the performing of back propagation from the beaching location.

Since back-propagation from the beaching location using HF radar currents is consistent with survivor's trajectory at the beginning and at the end, we believe the reviewer is right: improving NEMO results would lead to an improvement of Lagrangian tracking.

As we now point out in the paper, we performed a 10 month verification on HF radar data with NEMO and it seems the model performance during the Scirocco was particularly weak since the biases in (u,v) currents are otherwise much lower. Verification of the NEMO model versus ten months of hourly HF radar currents (not shown in this paper) yields a bias in zonal velocity between 0 and -2.5 cm/s and a bias between +2.5 and -2.5 cm/s for meridional velocity. NEMO model underestimations during the limited period of this case study were however even larger: spatially averaged (over the HF domain) and temporaly averaged (over the period of the drift) NEMO biases amounted to -6.3 cm/s for zonal velocity and a bias of -9.2 cm/s for meridional velocity. NEMO setup therefore exhibited below-average performance during the period of interest. This issue with model performance during storm conditions was explicitly pointed out in the paper and will have to further addressed in a separate investigation.

As far as Fig 6 is concerned, thanks for pointing this out. Plotted quantities are drifts, and therefore Eulerian velocities (in case of currents) or OpenDrift downwind slopes x u10 (in case of winds). Units are m/s. No other normalization is used. All this is now explicitly stated in the Figure caption.
* * *
Anonymous Referee #2

General Comments

In this work the potential trajectory followed by a windsurfer lost at sea in the surroundings of the Gulf of Trieste during a storm that took place in October 2018 is simulated through a Lagrangian approach.
To this end, high resolution oceanic and atmospheric models are used as input for two Lagrangian tools: OpenDrift and FlowTrack. Despite using the same numerical integration scheme (Runge-Kutta 2nd order) they present some differences. Likely the most important is that Open Drift offers pre-calibrated coefficients for downwind and crosswind components more suitable to simulate the drift of a person at sea, specially if the wind drift is dominant. As a result, the final distribution of particles simulated by OpenDrift match better the (tentative) inferred trajectory reconstructed by the authors from an interview with the windsurfer.

My impression is that this paper is based on a compelling idea and that results can be potentially useful at regional and operational level. English is good and one key strength is that model data (especially ocean currents) have very high resolution. However, I feel that this version of the manuscript needs some reorganization, a more accurate description of methods, a more elaborated assessment of the different con tributions of wind/ocean currents/(waves?) terms to the simulated trajectories, and an improved treatment of uncertainties and search and rescue areas, before being considered suitable for publication.

AR: We thank the reviewer sincerely for taking the time to comment on the paper!

In the below lines I elaborate further my above overall impression.

Major Comments
A- It is rather weird (and a bit confusing) to describe wave data from ECWAM model when it is not used later in the Lagrangian simulations. Indeed I think it is more logic to start in Section 2 with the quite general equations of Lagrangian particle tracking (current Section 4), and later describe in Section 3 observations and model data. In this way the fact that you neglect the Stokes drift is clearly stated and there is no need to include a description of the wave model in Section 3. Then in Section 4 you could show the results on the validation of model data with observations, and so on.

AR: We thank the reviewer for the comment. We agree that the approach was confusing. Before we clarify changes to the revised paper any further, we would like to say the following in our defense: it is true that ECWAM was not used in the final simulations with FlowTrack. But we did need ECWAM to compute the surface gravity wave wavelengths and to justify that we may indeed neglect Stokes drift. So we needed ECWAM to decide if we need it or not. This is why it was eventually presented. Since FlowTrack is out of the revised manuscript, ECWAM is also out of the paper. With OpenDrift this was not an issue since its parametrizations already include wave effects and OpenDrift consequently does not require the wave model at all. We also hope the existing section order is now working better.

Additionally, OpenDrift is now presented in more depth and additional simulations were made. We will address these changes point by point below.

B- You say that you do not consider the Stokes drift for your simulation because wavelengths are significantly larger than the size of a person/windsurf table. However, as far as I understand in the case of microplastics, wavelengths are proportionally even larger but the Stokes drift has an important role on their distribution (e.g. van den Bremer and Breivik; Onink et al., 2019). Am I wrong? I think that it is likely simpler to say that the Stokes drift is generally a second (or third) order term in the Adriatic Sea in terms of magnitude (e.g. see Fig. 1d within Onink et al., 2019) . Indeed it would be a good exercise to verify this point with an in-situ wave buoy or with your wave model data.

AR: As pointed out above, we now leave sections of FlowTrack, and consequently ECWAM, out of the revised paper. However, we will try to respond to the reviewer's question. Stokes drift is computed as a temporal mean of Lagrangian water particle velocity over its unclosed orbit in a surface gravity wave. It is in principle computed for the water particle moving passively with the wave field, not for material objects in the water. Although this is not a specialty of any of the authors, we would expect that the microplastics can safely be said to move directly with the fluid in the wave field. On the other hand, as far as we understand, there seems to be a difficulty to claim the same for larger material objects. Their orbits in the wave field do not generally follow closely the fluid motion due to a gravity wave. Therefore the temporal mean of the *object's* Lagrangian velocity over one cycle of its orbit, i.e. its "Stokes" drift, may differ substantially from the temporal mean of the surrounding *fluid's* Lagrangian velocity, i.e. water's Stokes drift. The argument was made in the references cited in the original version of the paper (Hackett et al. 2006, Breivik and Allen 2008) that we can only assume that object's orbit coincides to any significant degree with surrounding fluid particles orbits if the length of the material object is close to the surface gravity wave wavelength. In this case, Stokes drift calculation for fluid is a good estimate for the temporal mean of the object's unclosed orbits in the wave field. In other cases the orbits differ too much to make this assumption. This is how we understand the issue.

**C- It is not explained which expression you use to simulate turbulent effects (random numbers) at each time step. Is it based on a uniform or a normal distribution? Is it different for each direction? Is the maximum number of 0.02 m/s based on observations or a model assessment? Indeed in coastal regions where strong gradients in ocean currents exist the magnitude of diffusivity can change significantly between relatively close grid cells depending on the ocean features. Please clarify this point.**

AR: As noted above, FlowTrack was removed from the paper. Turbulent component was however computed from a normal distribution around this value (for each direction) which was inherited from an older Lagrangian code (unpublished), which based it from empirical vertical velocity shear data from some specific nice-weather situations (unpublished). We are aware that a constant turbulent diffusion is unacceptable for velocity fields with significant shear and we are improving FlowTrack to be able to ingest NEMO horizontal eddy viscosities, obtained via Smagorinsky or other schemes which take into account local velocity shear.

**D- With respect to the validation of the model data with observations, why do not show a comparison between the ADCP data (Figure 6, violet arrows) and the closest grid cell ocean model velocity? This would help to provide some numbers on the discussed underestimation of modeled currents. Also, do you have any reference in which HF radar velocities have been validated? Add it (them) to Section 2.2.**

AR: We thank the reviewer for the suggestion. We have followed it to perform a 10-month verification of the NEMO model against surface current HF radar data. We hope the reviewer will agree that this is a more extensive way to verify NEMO than a single-point ADCP comparison. We do not show plots of the results but we do state numerical values of the errors in the updated revision of the paper: they unfortunately indicate below-average model performance during the period of the drift and this is clearly pointed out in the paper.

The following passage was added to the paper: "Verification of the NEMO model versus ten months of hourly HF radar currents (not shown in detail in this paper) yields a bias in zonal velocity between 0 and -2.5 cm/s and a bias between +2.5 and -2.5 cm/s for meridional velocity. NEMO model underestimations during the limited period of this case study were however even larger: spatially averaged (over the HF domain) and temporaly averaged (over the period of the drift) NEMO biases amounted to -6.3 cm/s for zonal velocity and a bias of -9.2 cm/s for meridional velocity. NEMO setup therefore exhibited below-average performance during the period of interest. This will have to be further addressed as a separate issue and needs to be kept in mind when interpreting results below.
"

**E- Other unclear points are:**
**- How do you estimate the light red circles in Fig. 8-10?**
AR: We thank the reviewer for this question. These were initial estimates by the survivor. We now expand on our reply to the other reviewer in this regard. The survivor was not in posession of a GPS or a watch. Therefore both reviewers have justification to point out the arbitrary nature of his estimates. They are indeed arbitrary but we nevertheless took them as a rough estimate – this reviewer's comment has - in conjunction with demands from the second reviewer – led to our independently verification of the survivor's drift trajectory. This was done by interpolating in space and time the HF radar currents during the period of the drift, and to then use these measurements to compute Lagrangian *back*-propagation from the *beaching* location. This simulation yields a trajectory which seems largely consistent with survivors estimate. We have explained trajectory verification in a separate section and plotted the results in Figure 4. We hope the reviewer will agree that this substantiation gives some credibility to the survivor's estimate and that we can use it as a valuable qualitative orientation also during the drift, not only at the beaching point.

**- Why particles are initially deployed in a rectangular shape?**
AR: The square shape was simply one of the ways particles are seeded in FlowTrack. We comment a bit more below.

**- You deploy initially 480 particles, one for each value of Lp(theta), however in this way the uncertainty introduced by their different initial location is not assessed. What is the impact of the uncertainty in the initial position on the final search and rescue area for each Lp(theta)? Is it significant?**
AR: We thank the reviewer for this question. Yes, this is a good point. We haven't tested this sensitivity specifically but it is perhaps worth noting that the model resolution is 1 km and the deployment square size is 1 km. Therefore the dimension of the deployment shape is equal to the model grid size. Therefore we would expect that in this case the uncertainty in the initial position would not have, in itself, significant impact on the SAR areas since the model velocities change very little over the dimension of the release shape. However, note again that FlowTrack passages were removed from the paper.

**F – I find that would be interesting to show also the distribution of simulated particles with only wind drift/only ocean currents, and to estimate how large is the dispersion of particles (final area of search and rescue) for both cases. It would show graphically how predominant is the wind drift in the advection of particles and the lag with only ocean current.**

AR: We thank the reviewer for the suggestion: we have performed both (wind-only / current only) simulations for "person with surf board" drifter type as the reviewer suggested. We include the results below. Wind-only particle simulations were also included in the revised Figure 9 and in Figure 10 and we comment upon them in the text as well. Given that the situation was wind dominated, as is clear from "wind+current" and "wind-only" simulations, we did not include "current only" simulations in the paper.

Wind-only particle distribution after last time step:

[Figure]

Current-only particle distribution after last time step:

[Figure]

Wind+current distribution at the last timestep:

[Figure]

These figures indicate that winds are the dominant factor in this drift, but the currents play their role in elongating the spread along the coastal current "jet" direction (seen in left panel of Figure 6) thus moving it closer towards the beaching site.

**G- My major criticism to this work is the approach you follow to show your results. Considering that it is aimed to be useful for search and rescue (SAR) tasks and, therefore, time is critical, I think that to show trajectories is far less useful than to show areas (or contours) of accumulated probability constructed with suitable mathematical functions. In terms of accumulated probability the areas for search and rescue tasks can be more easily prioritized. Your estimated areas after just few hours of simulation look pretty large to be useful for search and rescue tasks. Additionally, the probabilistic approach naturally includes the fact that there are uncertainties everywhere. Even more, probability contours can have a bimodal distribution, while your polygon seems to include all particles inside irrespective of the spatial holes among them. For example, this approach can be found in Abascal et al., (2010). You need to convince me that for SAR tasks your current approach is reasonable enough, otherwise I suggest to redo your Fig. 8-10 with a probabilistic perspective (showing e.g. contours of 50%/70%/90% of accumulated probability estimated from the distribution of your particles), which is relatively easy to implement. In the latter case I suggest to remove the word "trajectory" from the title.**

AR: We thank the reviewer for the suggestion: we have rerun and replotted all simulations with OpenDrift to show numerical particle densities [number of particles / m2] which are a proxy for the probability of finding the particle on a given location at a given time (i.e. number of particles in a cell / number of released particles). Particle spread of modeled drifter types is in our case however very large and leads to very small probabilities on the scale of 1e-4 to 1e-5. We decided such numbers are not very telling and that particle density maps perhaps seem more intuitive to the reader. This density approach has been commonly used by the authors of OpenDrift themselves (Roehrs et al, 2018; Dugstad et al, 2019). We hope this satisfies the reviewer. We have followed reviewer suggestion and removed the word "trajectory" from the title.

**Other Comments**
Title. Change "for" by "of".
AR: Done.

**Ln 2. Suggest to change "He was drifting" by "He drifted".**
AR: Done.

**Ln 6. We "modeled".**
AR: This is now out of the text.

**Ln 52. Unclear how you estimate the +/- 500 m of error.**
AR: As noted, this is survivor's subjective estimate. We now state this in the paper explicitly and we add independent verification of survivor's trajectory using back-propagation on HF radar currents.

**Ln 65. "By the time he is entering" . . .**
AR: Done.

**Ln. 218. Remove point after "day"**
AR: Done.

**Ln. 218. "all directions..."**
AR: Done.

**Ln. 224. "generates a westward initial current".**
AR: Changed to "generates a westward inertial current"

[revised manuscript text omitted]

---

## Author Response (AR3)

Dear Editor, thank you for your comments. They have all been addressed in the final manuscript as shown below. Editor's review in gray, authors' response in black.

Dear Authors.
Thanks for having addressed the comments of the reviewers in your reply.

I am asking you a final minor revision, considering the two following points:

a) Considering the Stokes' drif (comment 1 of reviewer #2)

Line 178 I suggest adding "of the local wind sea" so that the sentence become "They already contain the Stokes drift contribution of the local wind sea in the observed leeway"

Thank you, corrected.

Lines 178-180 I suggest to replace
"If one attempts to model object's leeway using downwind and crosswind leeway coefficients based on empirical data from Allen and Plourde (1999), Stokes drift term must be omitted from the initial value problem (1)-(2), as explained in Breivik and Allen (2008)."
With
"In our attempt to model object's leeway using downwind and crosswind leeway coefficients based on empirical data from Allen and Plourde (1999), we have omitted the Stokes drift term from the initial value problem (1)-(2), as explained in Breivik and Allen (2008)."

Thank you, corrected.

b) Considering the accuracy of the NEMO fields and the comparison with HF observations (omment 3) of reviewer #2)

I appreciate your answer to comment 3) of reviewer # 2. I agree on your decision to keep the paper focused. Anyway, I think that you should expand a bit the discussion of the role of NEMO model errors and to which extent these errors might confuse your analysis of the tracking model. In my understanding, the two new experiments that you have performed show that errors on NEMO were large in the last 6 hours, but not previously. Please confirm this.

Thank you for the comment. We can confirm that NEMO performance was the worst at the end of the drift. We now quantify this in the text by stating the biases for the first 18 hours and for the last 6 hours of the drift. We have prepared a Supplementary Material pdf with these two graphs and a short description of the simulations. A reference is made in the text to supplementary material. The following text was added to Section 4.2:

"NEMO model underestimations during the limited period of this case study were unfortunately much larger: spatially averaged (over the HF domain) and temporaly averaged (over the period of the drift) NEMO biases amounted to -8.3 cm/s for zonal velocity and a bias of -8.8 cm/s for meridional velocity. Focusing into the period of the drift, NEMO model performance was best in the beginning and worst at the end of the drift. Over the first 18 hours of the drift, NEMO model exhibited -8.2 (8.1) cm/s bias in zonal (meridional) velocity, while during the last 6 hours of the drift these biases amounted to -11.4 (-11.1) cm/s. Additional forward-tracking simulations were performed to compare the advection due to HF and NEMO currents. These simulations are not included here but are described and depicted in Figure S2 in the supplementary material. They indicate that most of the error due to NEMO current underestimation was accumulated in the final hours of the drift."

I suggest that you add some text in section 4.1, mentioning these two experiments, the results of the back and forward tracking simulations and the conclusion that you get from their analysis. Consider adding a pdf with the maps as supplementary material (that can be cited in the text).

Thank you, implemented. The following text was included in section 4.1:

"Additional back-propagation simulation was performed with NEMO modelled currents in identical fashion. It allows comparison with back-propagation due to the HF currents. This simulation is not shown here but is depicted in Figure S1 in supplementary material. It indicates that NEMO currents tended to underestimate observed currents throughout the drift. This underestimation is however most notable in the final hours of the drift, as commented also below in Section \ref{Sec::MarineCond}."

Line 13 I suggest replacing "during the period of the drift" with "during the last 6 hours of the drift". This is my understanding of your answer to reviewer #2 comment 3

Thank you, corrected.

Line 14-15 the importance for what? My suggestion "This indicates the importance of topographically constrained ocean currents in semi-enclosed basins even in seemingly wind-dominated situations for determining the trajectory of a person lost at sea."

Thank you, corrected.

Looking forward to receiving the final version of your manuscript
Best regards
Piero Lionello

Supplementary material to manuscript:

nhess-2019-362
**Lagrangian Modelling of a Person lost at Sea during Adriatic Scirocco Storm of 29 October 2018**
Matjaž Ličer, Solène Estival, Catalina Reyes-Suarez, Davide Deponte, and Anja Fettich

**Backtracking from the beaching location with both HF radar and NEMO currents**

These simulations were performed as described in the body of the paper, Section 4.1. The result is shown on the image S1 below – left column depicts HF radar backpropagation, right column depicts NEMO backpropagation:

[Figure]

Figure S1.

During the final hours of the drift, NEMO currents seem to advect substantially differently (and attain less spatial distance from the beaching location) than the observed currents – hence the difference between the two "final" positions (locations of the red circles in Figure S1).

**Forward tracking from the beaching location with both HF radar and NEMO currents**

As pointed out in the paper, HF radar observational domain is limited in space. The accident occurred well outside the observational domain. Therefore we cannot use HF surface currents, as they are, to perform a forward tracking simulation of the entire drift. We can however attempt to perform such a simulation if we extrapolate the HF surface current field beyond its observational domain. While this is not entirely unproblematic, one can reasonably expect that such an extrapolation would be less questionable during a Scirocco than during any Bora episode. The reason for this distinction is that during the Scirocco the dominant surface currents along the northern Istrian coast flow northward, similarly as in the southern part of the mouth of the Gulf of Trieste. During Bora, the circulation in the southern part of the mouth of the Gulf can be however very different from the circulation along the northern Istrian coast. Southern part of the Gulf of Trieste during Bora typically exhibits a pronounced zonal flow while the circulation along the northern Istrian shore tends to be meridional due to the formation of a Bora driven double gyre system.

We have therefore performed a nearest-neighbor extrapolation of the WERA HF surface currents during the period of the drift. We have thus extended the HF surface current field to include the entire region of the drift. We have, to be clear, thus introduced the error which lies in the assumption that the currents in the vicinity of the accident are similar to those at the closest point of the radar observational domain. We then used this extrapolated field to perform forward tracking simulation of the Person with surfboard object type. The result is presented in Figure S2 below.

It is interesting to note that NEMO and HF currents tend to advect quite similarly until the final hours of the drift. The lag between the NEMO and the HF advected particles seems to be accumulated mostly in the last hours of the drift (again, the red circles in Figure S2).

[Figure]

Figure S2.

[revised manuscript text omitted]